# Acetylated tubulin is essential for touch sensation in mice

**Shane J Morley[1,2†], Yanmei Qi[3†], Loredana Iovino[1,2], Laura Andolfi[4], Da Guo[3], Nereo Kalebic[1,5], Laura Castaldi[1], Christian Tischer[6], Carla Portulano[1], Giulia Bolasco[1], Kalyanee Shirlekar[1], Claudia M Fusco[1], Antonino Asaro[1], Federica Fermani[1], Mayya Sundukova[1], Ulf Matti[6], Luc Reymond[7], Adele De Ninno[8], Luca Businaro[8], Kai Johnsson[7], Marco Lazzarino[4], Jonas Ries[6], Yannick Schwab[6], Jing Hu[3*], Paul A Heppenstall[1,2*]**

[1]EMBL Mouse Biology Unit, Monterotondo, Italy; [2]Molecular Medicine Partnership Unit (MMPU), Heidelberg, Germany; [3]Centre for Integrative Neuroscience, Tuebingen, Germany; [4]Istituto Officina dei Materiali-CNR, Trieste, Italy; [5]Max Planck Institute of Molecular Cell Biology and Genetics, Dresden, Germany; [6]European Molecular Biology Laboratory, Heidelberg, Germany; [7]Ecole Polytechnique Federale de Lausanne, Lausanne, Switzerland; [8]Consiglio Nazionale delle Ricerche, Rome, Italy

**\*For correspondence:** jing.hu@cin.uni-tuebingen.de (JH); paul.heppenstall@embl.it (PAH)

[†]These authors contributed equally to this work

**Competing interests:** The authors declare that no competing interests exist.

**Abstract** At its most fundamental level, touch sensation requires the translation of mechanical energy into mechanosensitive ion channel opening, thereby generating electro-chemical signals. Our understanding of this process, especially how the cytoskeleton influences it, remains unknown. Here we demonstrate that mice lacking the α-tubulin acetyltransferase Atat1 in sensory neurons display profound deficits in their ability to detect mechanical stimuli. We show that all cutaneous afferent subtypes, including nociceptors have strongly reduced mechanosensitivity upon Atat1 deletion, and that consequently, mice are largely insensitive to mechanical touch and pain. We establish that this broad loss of mechanosensitivity is dependent upon the acetyltransferase activity of Atat1, which when absent leads to a decrease in cellular elasticity. By mimicking α-tubulin acetylation genetically, we show both cellular rigidity and mechanosensitivity can be restored in Atat1 deficient sensory neurons. Hence, our results indicate that by influencing cellular stiffness, α-tubulin acetylation sets the force required for touch.

## Introduction

Mechanical forces acting upon cells or tissues are propagated into the opening of mechanically gated ion channels such as Piezo2 as the first step in the sense of touch (*Abraira and Ginty, 2013*; *Maksimovic et al., 2014*; *Ranade et al., 2014*; *Woo et al., 2014*). In many cases this process occurs through direct interplay of ion channels with the lipid bilayer. For example, the bacterial mechano-transduction channel MscS (*Sukharev, 2002*) and eukaryotic two-pore-domain potassium channels TRAAK and TREK1 (*Brohawn et al., 2014a*, *2012*, *2014b*; *Lolicato et al., 2014*) are fully activated by mechanical stimuli when reconstituted in reduced membrane systems, and the mechanosensitive ion channel Piezo1 is also likely to be gated by force exerted via lipids due to its exceptional sensitivity to membrane tension (*Cox et al., 2016*; *Lewis and Grandl, 2015*). However, in-vivo, membrane ion channels are not isolated from the cytoplasm and extracellular matrix, and mechanical sensitivity may depend on further interaction with other cellular components such as the underlying cytoskeleton to modify and redistribute membrane tension (*Delmas et al., 2011*; *Krieg et al., 2015*; *Qi et al., 2015*). Indeed, in *Drosophila*, NompC ion channels were shown to be linked between the

plasma membrane and microtubules by a tether protein domain in the N-terminus of the channel, and this linkage is essential for mechanosensitivity of the channel (*Zhang et al., 2015*). Moreover, in *C. elegans* touch sensitivity of specialized touch receptor neurons is dependent on both the actin binding protein β spectrin (*Krieg et al., 2014*) and the microtubule cytoskeleton (*Bounoutas et al., 2009*).

A feature of *C. elegans* touch receptor neurons is that their axons are filled with specialized cross-linked bundles of heavily acetylated 15-protofilament microtubules (*Chalfie and Thomson, 1982*). Disruption of the molecular components of these microtubules, MEC7 β-tubulin and MEC12 α-tubulin leads to a loss of mechanical sensitivity (*Bounoutas et al., 2009*; *Fukushige et al., 1999*). Moreover, mutation of MEC17, the major tubulin acetyltransferase (*Akella et al., 2010*; *Shida et al., 2010*) also reduces touch sensitivity in *C. elegans* (*Cueva et al., 2012*; *Topalidou et al., 2012*; *Zhang et al., 2002*). Of note, it is not clear whether this loss of touch sensitivity stems from the absence of tubulin acetylation in MEC17 mutants (*Shida et al., 2010*) or from other unknown actions of MEC17 (*Akella et al., 2010*; *Davenport et al., 2014*; *Fukushige et al., 1999*; *Topalidou et al., 2012*).

MEC17 functions to transfer an acetyl group to the lysine 40 (K40) residue on the luminal side of microtubules (*Szyk et al., 2014*), and this post-translation modification is remarkably well conserved in organisms that have cells with cilia (*Shida et al., 2010*). Intriguingly, Atat1, the mammalian ortho-logue of MEC17, is expressed ubiquitously in all mouse peripheral sensory neurons (*Kalebic et al., 2013a*), and these neurons have amongst the highest level of α-tubulin acetylation in the mouse (*Kalebic et al., 2013b*). This raises the question as to whether acetylated microtubules also have an essential function in mouse sensory neurons, and if yes, how, mechanistically they influence mechanosensation.

In this study, we investigated the contribution of microtubule acetylation to mammalian mechano-sensation by conditionally deleting Atat1 from mouse peripheral sensory neurons. We found that Atat1[cKO] mice display a profound loss of mechanical sensitivity to both light touch and painful stimuli with no impact on other sensory modalities. We demonstrate that this arises from a reduction in mechanosensitivity of all cutaneous afferent subtypes, including nociceptors, and adecreased mechanically activated currents in sensory neurons upon Atat1 deletion. We further establish that this broad loss of mechanosensitivity is dependent upon the acetyltransferase activity of Atat1, and that by mimicking α-tubulin acetylation genetically, mechanosensitivity can be restored in Atat1 defi-cient sensory neurons. Finally we show that acetylated microtubules localize to a prominent band under the membrane of sensory neuron cell bodies and axons, and in the absence of Atat1 and acet-ylated α-tubulin, cultured sensory neurons display significant reductions in their cell elasticity. Our results indicate that the microtubule cytoskeleton is an essential component of the mammalian mechanotransduction complex and that by influencing cellular stiffness, α-tubulin acetylation can tune mechanical sensitivity across the full range of mechanoreceptor subtypes.

## Results

### Atat1[cKO] mice display reduced sensitivity to innocuous touch and pain

To investigate cell autonomous effects of Atat1 disruption in sensory neurons we took a conditional gene deletion strategy. *Atat1*[fl/+] mice (*Kalebic et al., 2013b*) were crossed with a sensory neuron specific Cre driver line *Avil*-Cre (*Zurborg et al., 2011*) to generate *Avil*-Cre::*Atat1*[fl/fl] (referred to as Atat1[cKO]) and control *Avil*-Cre::*Atat1*[fl/+] mice (referred to as Atat1[Control]). Mice were then subjected to a series of behavioural assays. We first tested their ability to detect an innocuous mechanical stimulus applied to the hairy skin. Adhesive tape was fixed gently to the backs of animals and the number of responses counted over a 5 min observation period. While control mice made regular attempts to remove the tape, Atat1[cKO] mice effectively ignored the tape for much of the time, and the total number of responses was significantly lower ($p < 0.05$) (*Figure 1a*, *Video 1*). We next inves-tigated the sensitivity of mice to innocuous mechanical stimuli applied to the glabrous skin by lightly stroking the underside of the paw with a diffuse cotton swab. Again, Atat1[cKO] mice responded significantly less ($p < 0.01$) to this stimulus than Atat1[Control] mice (*Figure 1b*). We also examined whether mechanical sensitivity to punctate stimuli was altered in Atat1[cKO] mice by apply-ing von Frey filaments of calibrated forces to the hindpaw of mice. Control animals responded to

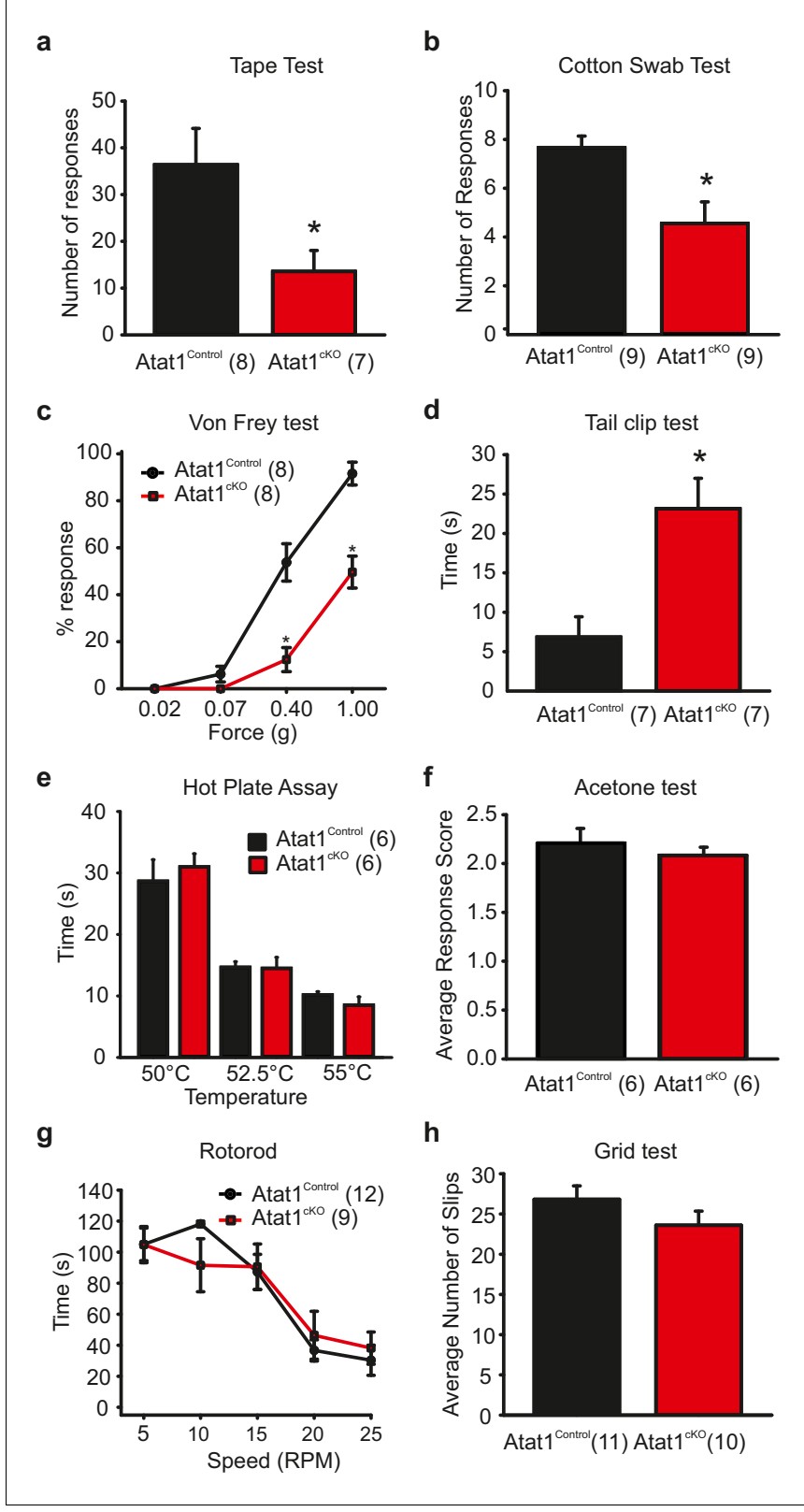

**Figure 1.** Behavioural analysis of Atat1cKOmice. (**a**) Bar-chart summarising the results of a tape test to assay low threshold mechanosensation. Atat1cKO mice demonstrated significantly less response events over the 5 min counting period (t-Test, p<0.05). (**b**) Results from the cotton swab analysis assaying low threshold mechanosensation. Atat1cKO mice demonstrated significantly less response events then Atat1Control counterparts

*Figure 1 continued on next page*

*Figure 1 continued*

(t-Test, p<0.01). (**c**) Graph of von Frey thresholds showing the significantly lower response frequency in Atat1[cKO] animals (Two-way RM ANOVA, Holm-Sidak method, p<0.001). (**d**) Bar-chart showing latency to a clip attached to the base of the tail. Atat1[cKO] animals take significantly longer to respond to the stimulus (t-Test, p<0.01). (**e**) No significant differences in the responses recorded to noxious heat between Atat1[cKO] and Atat1[Control] animals (t-Test, p>0.05). (**f**) No significant difference in cold response was observed using the acetone drop assay (t-Test, p>0.05) (**g**), No significant difference in motor performance as assayed using the Rotarod test (Two-Way RM ANOVA, Holm-Sidak method, p>0.05) and (**h**), No significant difference was observed in the average number of slips between the genotypes during the grid test (t-Test, p>0.05). Error bars indicate s.e.m.

The following figure supplement is available for figure 1:

**Figure supplement 1.** Behavioral analysis of Atat1[cKO] mice.

forces as low as 0.07g with an increase in detection into the noxious range. However, Atat1[cKO] mice required significantly higher forces to evoke a response throughout the range of von-Frey filaments (median threshold 0.4g for Atat1[Control] and 1g for Atat1[cKO] p=0.003, Mann-Whitney test) (*Figure 1c*). To investigate noxious mechanical sensitivity in more detail, we analysed responses to a clip applied to the base of the tail. Atat1[cKO] mice displayed substantially longer response latencies to the clip compared to Atat1[Control] mice and again, essentially ignored this noxious stimulus (p<0.01) (*Figure 1d*, *Video 2*). We further tested whether thermal detection was effected by Atat1 deletion by measuring responses to noxious heat or to evaporative cooling. We observed no difference in withdrawal latencies to a range of temperatures in the hotplate test between Atat1[cKO] and Atat1[Control] mice (*Figure 1e*) or to tail immersion in a hot water bath (*Figure 1—figure supplement 1*). Similarly, sensitivity to cooling of the paw with acetone was comparable in Atat1[cKO] and Atat1[Control] mice (*Figure 1f*). Finally we assessed the motor coordination and of Atat1[cKO] mice by evaluating their performance on a rotarod device or on a raised metal grid. Atat1[cKO] and Atat1[Control] mice displayed similar latencies to fall from the rotating drum across all speeds tested (*Figure 1g*) and a similar number of slips from the grid (*Figure 1h*). Thus, Atat1 is required for the detection of innocuous and noxious mechanical touch but not for noxious heat or proprioceptive coordination.

We further assessed the efficiency of Cre mediated recombination of the floxed *Atat1* allele in sensory neurons, and the suitability of using heterozygote *Avil*-Cre::*Atat1*[fl/+] mice as controls, by examining behavioural sensitivity to innocuous and noxious mechanical stimuli in full *Atat1* knockout mice (Atat1[-/-]) and wildype mice. In both the tape test and tail clip test, Atat1[cKO] mice performed similarly to Atat1[-/-] mice, and Atat1[Control] mice behaved comparably to wild-type (*Figure 1—figure supplement 1*). Thus in line with previous reports Avil-Cre mediated deletion is efficient and Cre expression has no apparent effect on sensory driven behaviour (*Zurborg et al., 2011*).

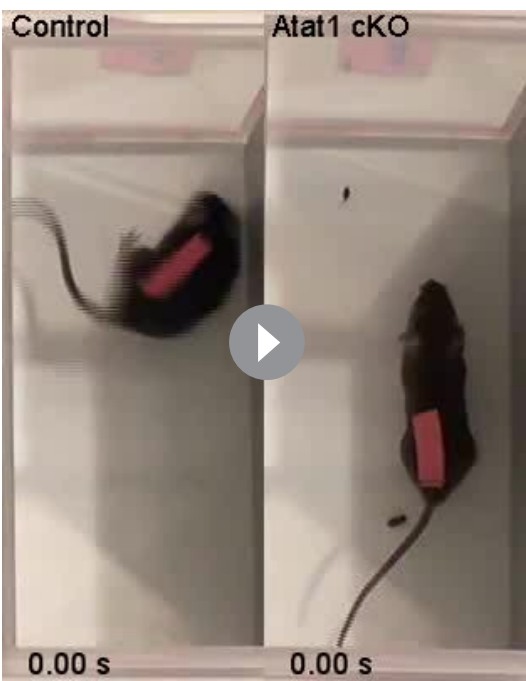

**Video 1.** Tape test assay. Video recording of mice showing attempts to remove a piece of tape by Atat1[Control] (left) and Atat1[cKO] (right), over a 5 min period.

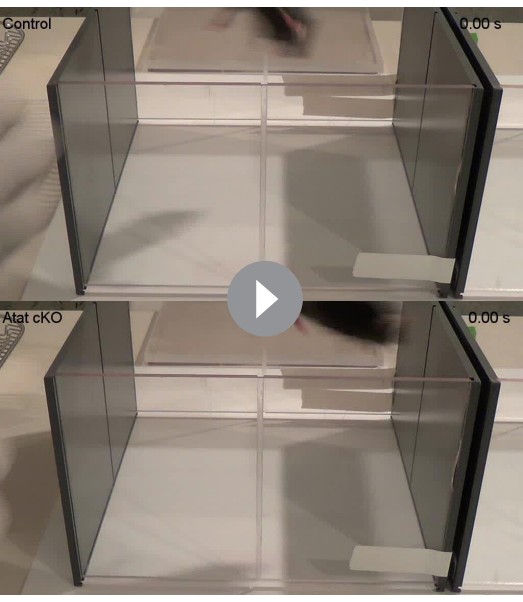

**Video 2.** Tail clip assay. Video of Atat1[Control] (top) and Atat1[cKO] (bottom) mice to measure the time taken until the first response of the animals to the clip.

# Atat1[cKO] mice display reduced mechanosensitivity across all mechanoreceptor subtypes innervating the skin

Sensory neuron axons terminate in the skin and form a diverse range of functionally distinct mechanoreceptors that underlie the sense of touch (*Abraira and Ginty, 2013*). They can be classified by their conduction velocity (into Aβ, Aδ and C fibres), their adaptation properties (into rapidly adapting or slowly adapting) and by their mechanical thresholds (into mechanoreceptors and mechanonociceptors). To determine whether the profound loss of mechanical sensitivity to both light touch and painful stimuli is due to the deficit in of each of these populations, we utilized an ex vivo skin-nerve preparation to record from single cutaneous sensory neurons in the saphenous nerve.

We first considered low threshold Aβ and Aδ fibres, separating them into slowly adapting (SAM) and rapidly adapting (RAM) Aβ mechanoreceptors, and Aδ D-hairs. We observed a striking reduction in the mechanical sensitivity of SAM fibres that was apparent as a reduced number of action potentials per stimulus indentation (*Figure 2a and b*, *Figure 2—figure supplement 1*) and a ~10-fold increase in the latency of the response (*Figure 2—figure supplement 1*) in Atat1[cKO] mice. Reductions in firing frequencies were evident during both the ramp phase (*Figure 2a*) of the mechanical stimulus and during the static phase (*Figure 2b*). RAM fibres displayed a similar reduction in their stimulus response function (*Figure 2c*, and *Figure 2—figure supplement 2*) and an increased latency to the highest displacement stimulus (*Figure 2—figure supplement 2*). A characteristic of these fibres is that they display higher firing frequencies with increasing stimulus speed (*Milenkovic et al., 2008*), a feature which was also reduced in Atat1[cKO] mice (*Figure 2d* and *Figure 2—figure supplement 2*). D-hairs also displayed significant reductions in their stimulus response function (*Figure 2e* and *Figure 2—figure supplement 3*), as well as longer latencies for mechanical activation (*Figure 2—figure supplement 3*), and decreased sensitivity to dynamic stimuli (*Figure 2f* and *Figure 2—figure supplement 3*) in Atat1[cKO] mice. Electrical thresholds and conduction velocities were unchanged in low threshold Aβ and Aδ fibres in the absence of Atat1 (*Figure 2—figure supplements 1–3*).

We next examined high threshold nociceptive fibres which can be classified by their conduction velocities into Aδ-mechanonociceptors (AM) and C-fibre mechanonociceptors. Similar to low threshold fibres, both populations of nociceptor exhibited a reduced number of action potentials evoked by indentation (AM units: *Figure 3a and b*, *Figure 3—figure supplements 1* and *2*), longer latencies for mechanical activation (*Figure 3—figure supplements 1* and *2*) and no change in electrical thresholds and conduction velocities in the absence of Atat1 (*Figure 3—figure supplements 1* and *2*). Mechanical thresholds to von Frey stimuli were also elevated in C-fibre nociceptors (*Figure 3—figure supplement 2*) highlighting the strength of the phenotype in this population. Finally, we analysed heat responsiveness of C-fibre nociceptors in Atat1[Control] and Atat1[cKO] mice. We observed no difference between genotypes in the proportion of C-fibres responding to heat, or in their activation thresholds and firing rate in response to a heat stimulus (*Figure 3—figure supplement 3*). Thus collectively, these data indicate that Atat1 is specifically required for mechanical sensitivity across all major fibre types innervating the skin.

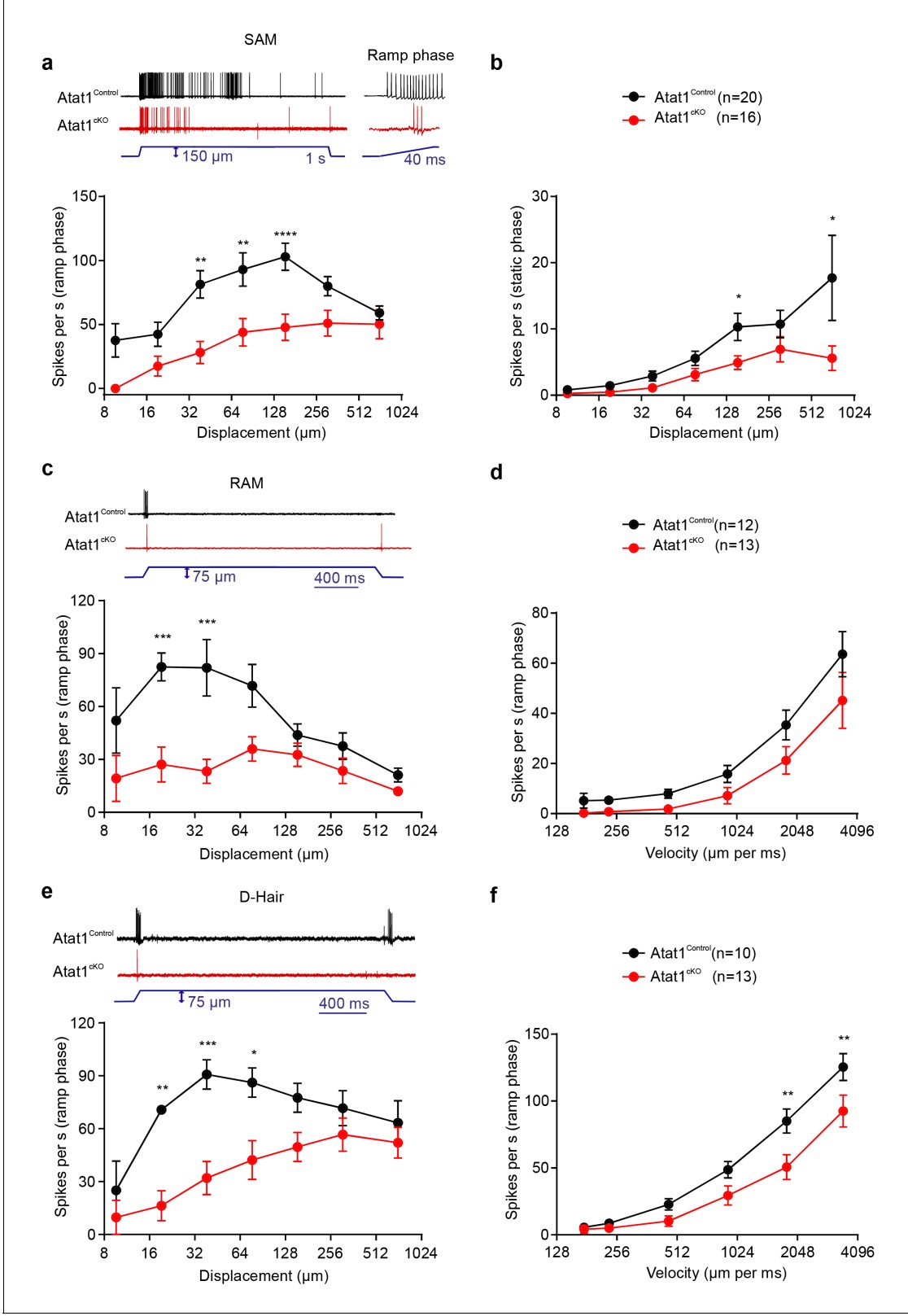

**Figure 2.** αTAT1 is required for mechanosensitivity of low threshold mechanoreceptors. Slowly adapting mechanoreceptor fibers (SAM): Typical responses (top) and stimulus-response functions (bottom) to increasing displacement for the ramp (**a**) and static (**b**) phase (two-way ANOVA with *post-hoc* Bonferroni's test, ramp phase: p<0.0001; static phase: p<0.001). Rapidly adapting mechanoreceptor fibers (RAM): Typical responses (top) and stimulus-response functions (bottom) to increasing displacement (**c**) and velocity (**d**) (two-way ANOVA with *post-hoc* Bonferroni's test, displacement:
*Figure 2 continued on next page*

*Figure 2 continued*

p<0.0001; velocity: p<0.001). D-hair afferents: Typical responses (top) and stimulus-response functions (bottom) to increasing displacement (**e**) and velocity (**f**) (two-way ANOVA with *post-hoc* Bonferroni's test, displacement: p<0.0001; velocity: p<0.0001). The number of fibres recorded is indicated in parentheses in each panel. *p<0.05; **p<0.01; ***p<0.001; ****p<0.0001. Error bars indicate s.e.m.

The following figure supplements are available for figure 2:

**Figure supplement 1.** Electrical excitability and stimulus response properties of SAM fibres in Atat1$^{Control}$ and Atat1$^{cKO}$ mice.

**Figure supplement 2.** Electrical excitability and stimulus response properties of RAM fibres in Atat1$^{Control}$ and Atat1$^{cKO}$ mice.

**Figure supplement 3.** Electrical excitability and stimulus response properties of D-hair fibres in Atat1$^{Control}$ and Atat1$^{cKO}$ mice.

## Morphological analysis of the peripheral nervous system in Atat1$^{cKO}$ mice

The absence of Atat1 could potentially have wide-ranging effects on the development, morphology and structure of peripheral sensory neurons, each of which could contribute to a loss of mechano-sensitivity. We thus performed a systematic analysis of the organization of the peripheral nervous system upon deletion of Atat1. We first tested whether the number of sensory neurons and their innervation of the skin and spinal cord were affected by deletion of Atat1. We observed no change in the number of myelinated and unmyelinated fibres in the saphenous nerve (*Figure 4a–d*), in the number, density, or organization of terminal endings in the skin (*Figure 4e–h*), or in the innervation of the spinal cord by CGRP, IB4 and NF200 positive sensory neurons (*Figure 4m–p* and *Figure 4— figure supplement 1*). We next examined axonal outgrowth and neurotrophin/receptor transport in sensory neurons. In whole mount DRG explants supported in the presence of NGF we detected no difference in the growth rate or length of axons in DRG from Atat1$^{Control}$ and Atat1$^{cKO}$ mice (*Figure 4i–k*). Moreover, single molecule imaging of NGF/receptor transport in a microfluidic device revealed that the displacement and velocity of quantum dot labelled NGF molecules (*Sung et al., 2011*), was not altered in the absence of Atat1 (*Figure 4l*, *Figure 3—figure supplement 2* and *Video 3*). Taken together with electrophysiological analysis, these data indicate that the loss of mechanosensitivity is not due to generalized effects on neuronal function but rather arises from defects in the ability of sensory neurons to transduce mechanical forces into electrical signals.

## Deficits in mechanically activated currents in DRG neurons from Atat1$^{cKO}$ mice

To determine how deletion of Atat1 influences mechanotransduction in sensory neurons we recorded mechanosensitive currents from cultured DRG neurons indented with a blunt glass probe. Such a stimulus can evoke mechanically gated currents in ~90% of DRG neurons that are further classified as rapidly adapting (RA), intermediate-adapting (IA) and slowly adapting (SA) responses (*Hu and Lewin, 2006*). In the absence of Atat1, we observed a marked loss in the number of mechanically sensitive neurons evoked by cell prodding that was evident across each subtype of current (*Figure 5a*). Furthermore, the small proportion of neurons which still displayed mechanosensitive currents in Atat1$^{cKO}$ mice exhibited significantly reduced current amplitudes and higher thresholds (*Figure 5b–d*, *Figure 5—figure supplement 1*), but no difference in their activation kinetics (*Figure 5—figure supplement 1*). Other functional parameters such as voltage gated channel activity, resting membrane potential, action potential threshold, and sensitivity to capsaicin or pH were indistinguishable between Atat1$^{Control}$ and Atat1$^{cKO}$ mice (*Figure 5—figure supplements 2* and *3*). We also examined perimembrane dynamics of the mechanosensitive ion channel Piezo2 in the absence of Atat1, and observed no difference in Fluorescence Recovery After Bleaching (FRAP) of a Piezo2-GFP fusion construct transfected into dissociated neurons (*Figure 5—figure supplement 4*). Thus the reduced mechanical sensitivity of DRG neurons in Atat1$^{cKO}$ mice does not arise from compromised membrane properties in these cells or defects in mechanosensitive ion channel trafficking.

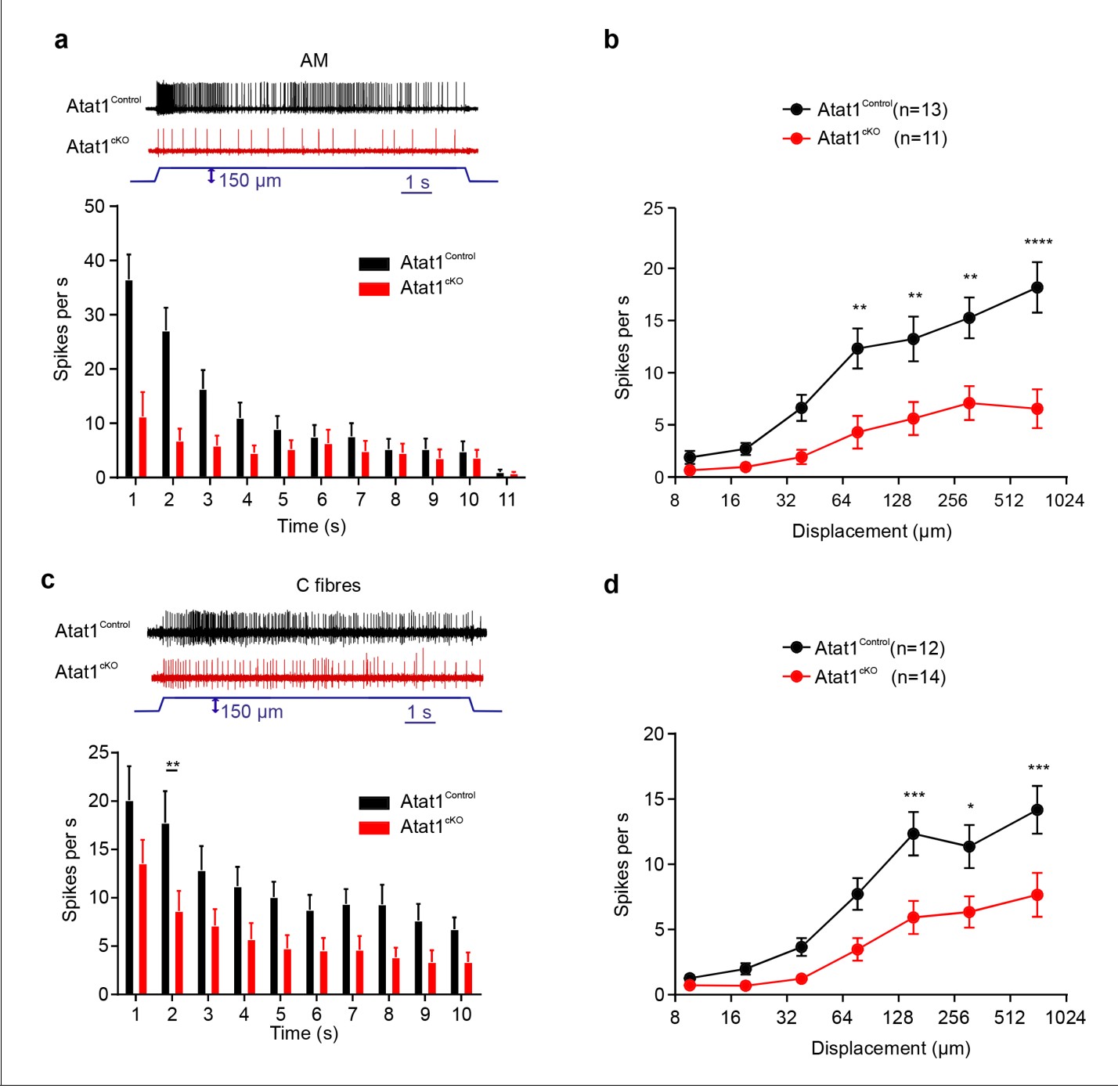

**Figure 3.** αTAT1 is required for mechanosensitivity of nociceptors. Typical responses (top) and mean discharge rates (1 s bins) during 10 s 150 μm stimulus of Aδ-mechanonociceptors (AM) (a) and C-fibre nociceptors (c) from $\alpha TAT1^{control}$ and $\alpha TAT1^{cko}$ mice (two-way ANOVA with *post-hoc* Bonferroni's test, AM: p<0.05; C-fibre: p<0.0001). (b) Stimulus-response functions (bottom) to increasing displacement for AM (b) and C-fibre nociceptors (d) (two-way ANOVA with *post-hoc* Bonferroni's test, AM: p<0.0001; C-fibre: p<0.0001). The number of fibres recorded is indicated in parentheses in each panel. *p<0.05; **p<0.01; ***p<0.001; ****p<0.0001. Error bars indicate s.e.m.

The following figure supplements are available for figure 3:

**Figure supplement 1.** Electrical excitability and stimulus response properties of AM fibres in Atat1[Control] and Atat1[cKO] mice.

**Figure supplement 2.** Electrical excitability and stimulus response properties of C fibres in Atat1[Control] and Atat1[cKO] mice.

*Figure 3 continued on next page*

*Figure 3 continued*

**Figure supplement 3.** Heat response of C fibers in Atat1[Control] and Atat1[cKO] mice.

## Genetically mimicking α-tubulin acetylation restores mechanosensitivity of Atat1[cKO] sensory neurons

We next asked whether the reduction in mechanosensitivity observed in Atat1[cKO] mice is dependent upon the α-tubulin acetyltransferase activity of Atat1 by testing if mechanically activated current properties could be re-established to control levels by expression of exogenous cDNAs. As a positive control we determined that transfection of an Atat1-YFP construct rescued mechanosensitivity in Atat1[cKO] cultures and that the proportion of RA, IA and SA responses across the DRG returned to control levels (*Figure 5e*). We subsequently transfected a catalytically inactive form of Atat1 (termed Atat1-GGL) that has no acetyltransferase activity but remains functional (*Kalebic et al., 2013a*). Expression of Atat1-GGL did not restore mechanosensitivity in Atat1[cKO] neurons, and we observed no difference in the proportion of different types of mechanically activated current compared to mock EGFP transfection (*Figure 5e*). Thus the acetyltransferase activity of Atat1 is required for normal mechanical sensitivity of DRG neurons.

Atat1 has also been demonstrated to acetylate other substrates in addition to α-tubulin (*Castro-Castro et al., 2012*). Therefore, to determine whether α-tubulin acetylation underlies the mechanosensory phenotype in Atat1[cKO] mice, we transfected a K40Q point mutant of α-tubulin that genetically mimics α-tubulin lysine 40 acetylation. Expression of K40Q α-tubulin rescued mechanosensitivity of Atat1[cKO] DRG neurons to Atat1[Control] levels, while a charge conserving control mutation (K40R) had no significant effect (*Figure 5f*). Collectively these data strongly indicate that the acetyltransferase activity of Atat1 modulates mechanosensitivity and that acetylated α-tubulin is the likely effector.

## Cytoskeletal organization in sensory neurons from Atat1[cKO] mice

A recent study has demonstrated that the mechanosensitivity of hypothalamic osmosensory neurons is dependent upon a unique interweaved organization of microtubules in these cells (*Prager-Khoutorsky et al., 2014*). We therefore asked whether a similar structure is also evident in peripheral sensory neurons and whether it is dependent upon the presence of acetylated α-tubulin. We performed superresolution dSTORM (direct stochastic optical reconstruction microscopy)(microscopy on DRG neurons stained with α-tubulin antibodies, and indeed observed that microtubules form an interweaved network in all neurons examined, especially towards the centre of the cell (*Figure 6a and b*). However, we were unable to detect clearly visible differences in organization between Atat1[Control] and Atat1[cKO] sensory neurons (*Figure 6a and b*). To assay this more quantitatively we subjected superresolution images to an unbiased automated analysis (*Figure 6c–f*). We observed no difference in microtubule density (*Figure 6g*), the number of microtubule crossing points (*Figure 6h*), or the angular variance of the microtubule cytoskeleton (*Figure 6i*) in any neuron between Atat1[Control] and Atat1[cKO] mice indicating that acetylation does not influence the gross organization of the microtubule network. To investigate whether the absence of microtubule acetylation could indirectly impact upon the actin cytoskeleton, we also performed phalloidin staining on DRG neurons. Again we did not detect any difference in the organization of the actin network upon deletion of Atat1 (*Figure 6—figure supplement 1*).

## Acetylated tubulin localizes to a prominent submembrane band in peripheral sensory neurons

We next investigated the contribution of acetylated tubulin to mechanosensitivity by examining the distribution of acetylated microtubules in sensory neurons. Strikingly, we observed that acetylated α-tubulin was concentrated in a band directly under the plasma membrane in cultured DRG neurons that was evident in 80 ± 5% of neurons (*Figure 7a* and *Figure 6—figure supplement 1*), while total α-tubulin was distributed evenly across the cytoplasm of all cells (*Figure 7b*). Importantly, this band was not present in non-mechanosensory cells such as serum-starved fibroblasts where acetylated α-

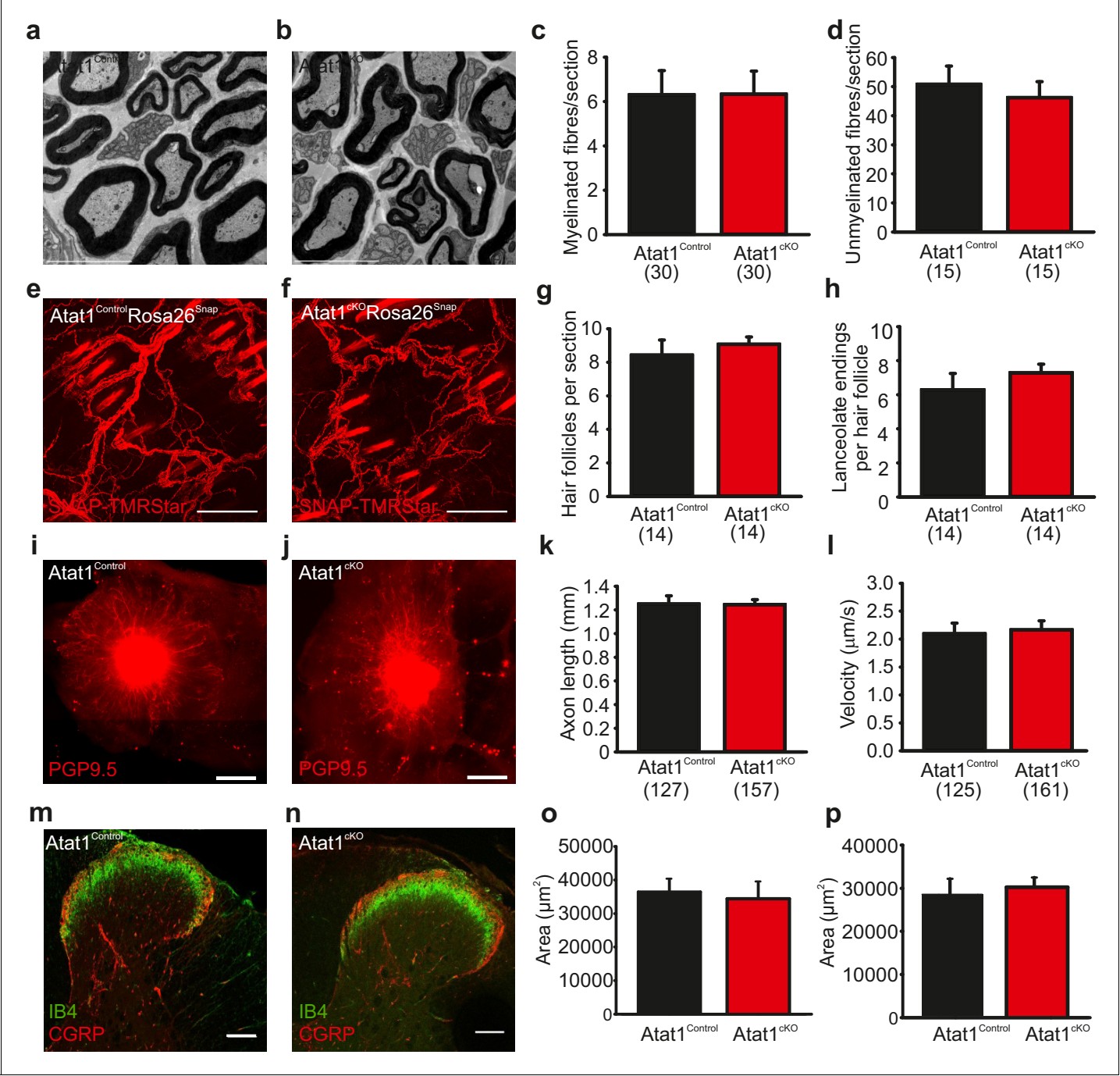

**Figure 4.** Morphological analysis of the peripheral nervous system in Atat1[cKO] mice. (**a** and **b**) Electron micrographs of a sectioned saphenous nerve from an Atat1[Control] (**a**) and Atat1[cKO] (**b**) mouse (Scale bar 5 μm). (**c** and **d**) Graph summarizing the number of myelinated fibres (**c**) and unmyelinated fibres (**d**) from saphenous nerve sections. No significant difference was noted between the samples (t-Test, p>0.05). (**e** and **f**) Fluorescent image of skin taken from the finger of an Atat1[Control]::Rosa26[Snap] mouse (**e**) and Atat1[cKO]::Rosa26[Snap] mouse (**f**) labelled with SNAP-TMR Star (Scale bar 100 μm). (**g** and **h**) Bar-chart showing the number of innervated hair follicles (**g**) and lanceolate endings per hair follicle (**h**) in skin taken from the digits of Atat1[Control]::Rosa26[Snap] and Atat1[cKO]::Rosa26[Snap] animals. No significant difference was observed between the genotypes (t-Test, p>0.05). (**i** and **j**) Mosaic image of PGP9.5 stained whole mount DRG after seven days in culture from Atat1[Control] (**i**) and Atat1[cKO] (**j**) mice (Scale bar 500 μm). (**k**) Bar-chart showing the length of axonal outgrowths after 7 days of culture. No significant difference was observed in Atat1[Control] and Atat1[cKO] DRG (t-Test, p>0.05). (**l**) Single molecule tracking of NGF molecules in neurites from Atat1[Control] and Atat1[cKO] DRG grown in microfluidic devices. Graph showing the average instantaneous velocity for NGF coupled quantum dots. No significant difference in NGF transport was observed between the genotypes (t-Test, p>0.05). (**m** and **n**) Confocal images of sectioned spinal cord dorsal horn from Atat1[Control] (**m**) and Atat1[cKO] (**n**) stained with IB4 (green) and CGRP

*Figure 4 continued on next page*

*Figure 4 continued*

(red) antibody respectively (Scale bar 100 µm). (**o**) Bar-chart showing the size of the area stained with IB4. No significant difference was observed in the size of the area between Atat1^Control^ and Atat1^cKO^ spinal cord (t-Test, p>0.05). (**p**) Bar-chart showing the size of the area stained with a CGRP antibody. No significant difference was observed in the size of the area between Atat1^Control^ and Atat1^cKO^ spinal cord (t-Test, p>0.05). Error bars indicate s.e.m.

The following figure supplements are available for figure 4:

**Figure supplement 1.** Spinal cord staining in Atat1^Control^ and Atat1^cKO^ mice.

**Figure supplement 2.** Axonal transport in DRG neurons from Atat1^cKO^ mice.

tubulin was found throughout the microtubule network (*Figure 7c and d*). We further examined the distribution of acetylated α-tubulin in intact preparations of the peripheral nervous system.Acetylation was enriched under the membrane of axons in the saphenous nerve (as determined from staining with a Myelin Basic Protein antibody) (*Figure 7—figure supplement 1*) and also apparently at sensory neuron terminal endings in the cornea where mechanotransduction takes place (*Figure 7—figure supplement 1*, *Video 4*).

## Increased rigidity of DRG neurons from Atat1^cKO^ mice

What then is the function of the acetylated α-tubulin band, and how does it impact upon mechanosensitivity across the range of mechanoreceptors in the skin? One possibility is that it sets the rigidity of cells thereby influencing the amount of force required to displace the plasma membrane and activate mechanosensitive channels. We explored this by directly measuring cell elasticity using atomic force microscopy. In DRG neurons from Atat1^cKO^ mice we observed that cellular stiffness was significantly higher across a range of indentations extending from displacements of 200 nm to 600 nm (p<0.01) (*Figure 7e* and *Figure 7—figure supplement 2*). Thus higher forces are required to indent sensory neurons from Atat1^cKO^ mice than Atat1^Control^ mice.

We confirmed this further by assaying the shrinkage of sensory neurons induced by a hyperosmotic stimulus. In the absence of Atat1, sensory neuron axons displayed less shrinkage than their control counterparts, an effect that could be rescued by expression of the acetylation mimicking mutation α-tubulin K40Q (p<0.05) (*Figure 7f*). Finally we examined how the microtubule cytoskeleton responds to compression induced by osmotic pressure. Using a novel tubulin labelling fluorescent dye we were able to resolve individual microtubule bundles in live imaging experiments. Strikingly, in DRG neurons from Atat1^cKO^ mice we observed significantly reduced microtubule displacement upon application of hyperosmotic solutions compared to control and K40Q expressing cells (p<0.05) (*Figure 7g* to m), again supporting the premise that in the absence of α-tubulin acetylation sensory neurons are more resistant to mechanical deformation.

## Discussion

Here we establish that deletion of the α-tubulin acetyltransferase Atat1 from mouse peripheral sensory neurons results in a profound and remarkably selective loss of cutaneous mechanical sensation. We demonstrate that this impacts strongly upon both light touch and pain, and that all mechanoreceptor subtypes which innervate the skin are less responsive in the absence of Atat1. This phenotype arises from reduction of the amplitude of mechanically activated currents in sensory neurons, and we propose that it is mediated by the loss of a sub-membrane band of acetylated α-tubulin in Atat1^cKO^ mice that sets mechanical rigidity of these cells. Our data thus

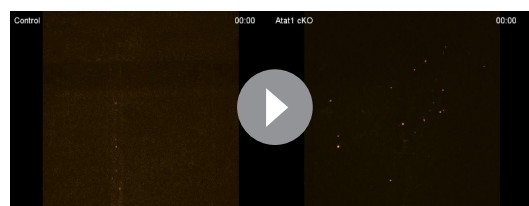

**Video 3.** Microfluidics. Movement of NGF labelled quantum dots along the neurites of cultured DRG. Atat1^Control^ (left) and Atat1^cKO^ (right).

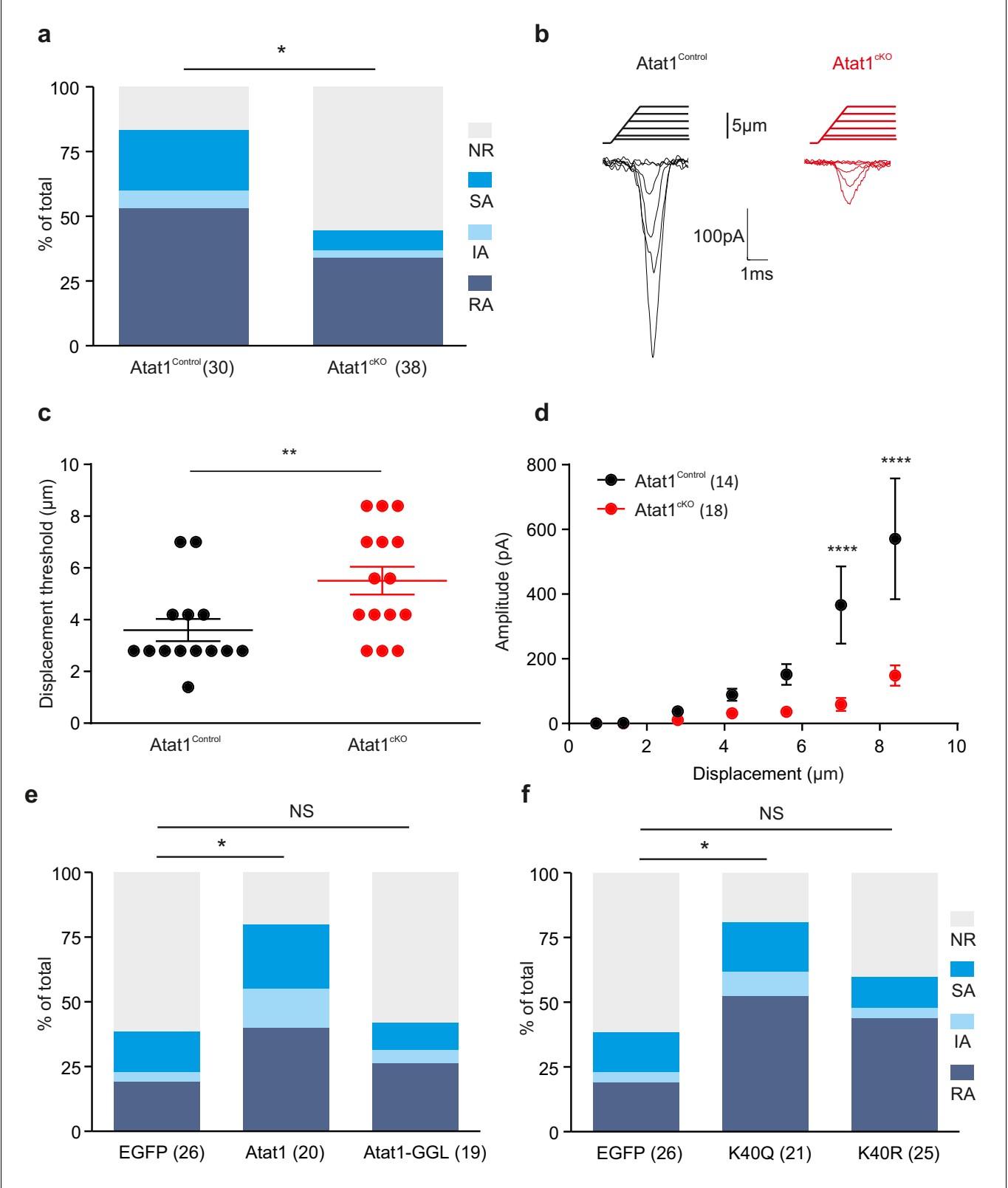

**Figure 5.** Atat1-mediated acetylated microtubules regulate mechanosensitivity in sensory neurons. (a) Stacked histograms showing the proportion of different mechano-gated currents activated by neurite indentation in sensory neurons from control Atat1[Control] and Atat1[cKO] mice ($\chi^2$ test, p<0.05). NR, non-responsive to given displacement 512 nm. (b) Representative traces of RA currents elicited by increasing probe displacement on soma of Atat1[Control] and Atat1[cKO] sensory neurons. (c) Threshold of activation of RA currents was determined as mechanical stimulus that elicited a current ≥ 20

*Figure 5 continued on next page*

*Figure 5 continued*

pA. Closed circles indicate individual recorded cells. Note the marked increase in the displacement threshold in Atat1[cKO] sensory neurons (Mann-Whitney test, p<0.01). (**d**) Stimulus-response curve of RA currents evoked by increasing probe displacement. Genetic depletion of aTAT1 in sensory neuron significantly reduced RA-currents amplitude (two-way ANOVA with *post-hoc* Bonferroni's test, p<0.0001). (**e**) Stacked histograms showing the proportions of different mechano-gated currents from stimulation of the neurites observed in Atat1[cKO] sensory neurons transfected with EGFP, Atat1-YFP or Atat1-GGL-YFP cDNA. Transfection of wild-type Atat1 rescued the loss of mechanosensitivity, while transfection of catalytically inactive Atat1 (Atat1-GGL-YFP) failed to restore it in Atat1[cKO] sensory neurons ($\chi^2$ test, EGFP versus Atat1-YFP, p<0.05; EGFP versus Atat1-GGL-YFP, p>0.05). (**f**) Stacked histograms showing the proportions of different mechano-gated currents from stimulation of the neurites observed in Atat1[cKO] sensory neurons transfected with *EGFP*, α-tubulin[K40R]-IRES-YFP (*K40R*) or α-tubulin[K40Q]-IRES-YFP (K40Q) cDNA. Transfection of acetylated α-tubulin mimics (K40Q) but not non-acetylatable α-tubulin mutant (K40R) restored mechanosensitivity in Atat1[cKO] sensory neurons ($\chi^2$ test, EGFP versus K40Q, p<0.05; EGFP versus K40R, p>0.05). The number of neurons recorded is indicated in parentheses in each panel. \*\*p<0.01; \*\*\*\*p<0.0001; Error bars indicate s.e.m.

The following figure supplements are available for figure 5:

**Figure supplement 1.** Absence of Atat1 in sensory neurons does not alter kinetic properties of RA currents.

**Figure supplement 2.** Absence of Atat1 in sensory neurons does not alter membrane electrophysiological properties and proton activated currents.

**Figure supplement 3.** Capsaicin response of DRG neurons in Atat1[Control] and Atat1[cKO] mice.

**Figure supplement 4.** FRAP analysis of Piezo2 in transfected DRG from Atat1[Control] and Atat1[cKO] mice.

describe a model whereby cellular stiffness regulates mechanical sensitivity by setting the force required to indent the plasma membrane and activate mechanosensitive ion channels.

In addition to its well characterized role as an α-tubulin acetyltransferase, Atat1 has been shown to have functions beyond its acetyltransferase activity at α-tubulin (*Castro-Castro et al., 2012*; *Kalebic et al., 2013a*). Indeed in C. elegans, several reports indicate that acetylated tubulin is not required for touch sensitivity (*Akella et al., 2010*; *Davenport et al., 2014*; *Fukushige et al., 1999*; *Topalidou et al., 2012*) and as such it is unknown as to how MEC17 (the C. elegans orthologue of Atat1) regulates mechanosensation in nematodes. It was therefore important to determine whether α-tubulin acetylation does indeed underlie the loss of mechanosensation in Atat1[cKO] mice. We addressed this through genetic rescue experiments where we attempted to restore mechanosensitivity of Atat1[cKO] sensory neurons by exogenous expression of Atat1 and α-tubulin point mutants. Importantly, we found that an acetyltransferase deficient form of Atat1 had no effect on mechanically activated current amplitude, while genetically mimicking tubulin acetylation with an α-tubulin K40Q mutation did restore mechanosensitivity. Thus acetylated microtubules are likely the effector for the loss of mechanosensation in Atat1[cKO] mice, suggesting that mammals and nematodes may utilize Atat1 differentially in controlling touch sensation.

Given the multitude of cellular processes that depend on microtubules, especially in cells with elongated axons such as peripheral sensory neurons, one possible mechanism by which Atat1 might be affecting mechanosensitivity is via altered structure of the peripheral nervous system, perhaps as a result of aberrant axonal transport in the absence of acetylated microtubules (*Reed et al., 2006*). We performed a thorough analysis of the organization of the peripheral sensory system and observed no differences in rates of neurotrophin/receptor transport and axon outgrowth in vitro, or in the ultrastructure of peripheral nerves and their innervation of the skin and spinal cord in vivo. Moreover, deletion of Atat1 was remarkably selective in that it impacted only on cutaneous mechanosensation and had no effect on thermal responses, or on electrical or other functional properties of sensory neurons such as their sensitivity to capsaicin and pH. We have previously reported that STOML3, another critical mediator of mechanotransduction, associates with microtubules in a specialized vesicle where it interacts with acid-sensing (proton-gated) ion channel (ASIC) subunits (*Lapatsina et al., 2012*). Uncoupling of vesicles from microtubules leads to incorporation of STOML3 into the plasma membrane and increased acid-gated currents. Acetylation status of microtubules could conceivably influence the transport of these vesicles and contribute to the phenotype in

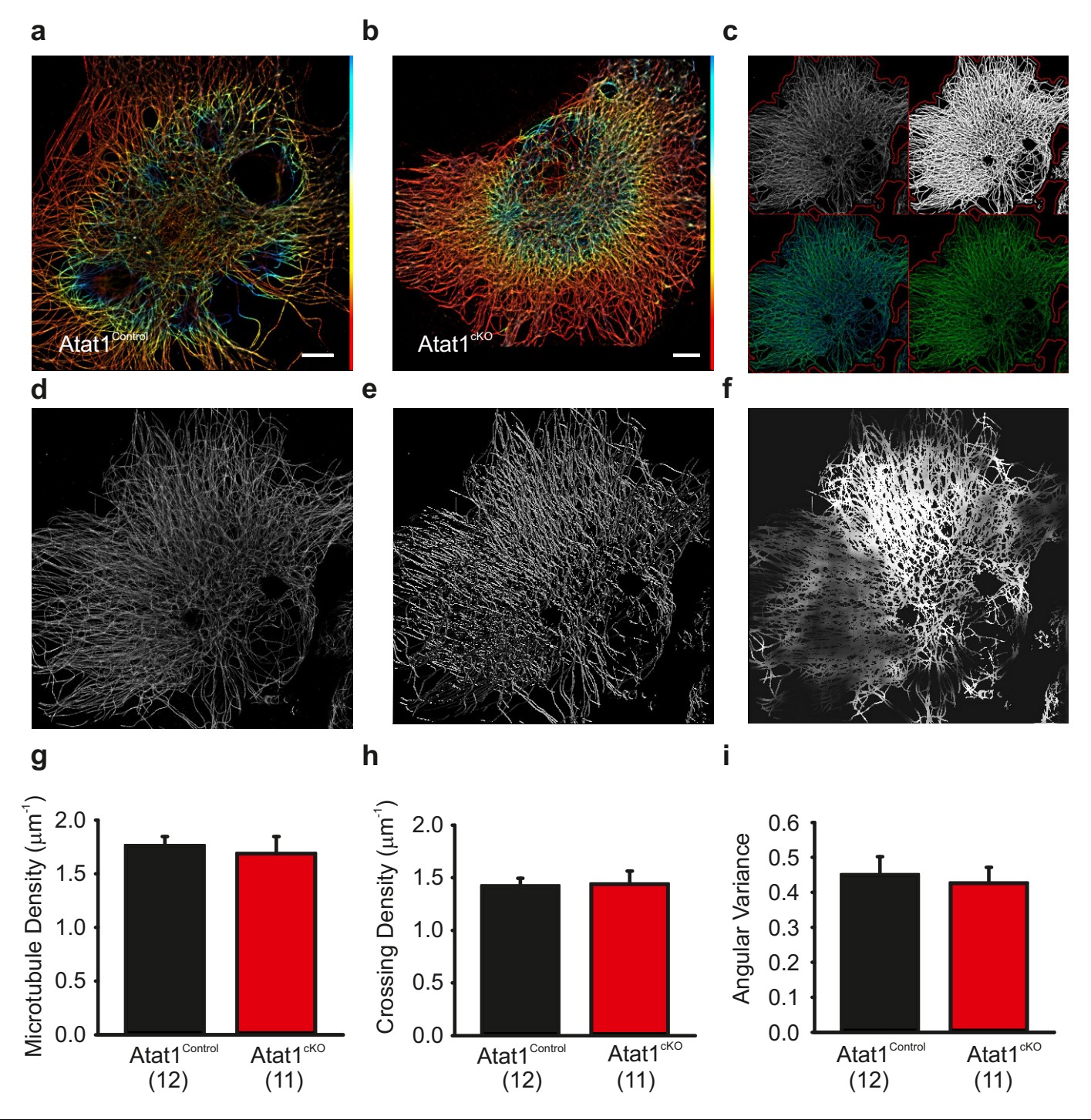

**Figure 6.** Superresolution imaging of microtubules in DRG from Atat1^Control and Atat1^cKO mice. (a and b) Superresolution image of anti α-tubulin staining of Atat1^Control (a) and Atat1^cKO (b) DRG colour coded by depth (red close to objective, Scale bar 5 μm). (c) Graphical representation of automated analyses performed on superresolution images. The top left image shows the original image in grey and the automated selection of the imaged cell area as a red outline. The top right image shows the microtubule image after binarization. The bottom left image shows the detected microtubule skeleton (in blue) overlaid on the microtubule superresolution image (in green); the microtubule skeleton was used to measure network length. The bottom right image is an estimation of the number of branch points present in the microtubule network, with microtubules in green and the branch points marked in blue. (d–f) Graphical representation of the automated analysis of the microtubule network angular variance. (d) Shows the microtubule superresolution image, (e) the automatically determined orientation of the microtubules, (f) the local angular variance of the microtubule orientation, where bright pixels denote a high variance and vice versa. (g) Graph summarizing the microtubule density in DRG taken from Atat1^Control

*Figure 6 continued on next page*

*Figure 6 continued*

and Atat1$^{cKO}$ mice. (h) Graph showing the density of microtubule crossings present in the microtubule networks in the two genotypes. (i) Graph summarizing the angular variance of the microtubule cytoskeleton in both Atat1Control and Atat1cKO cultured DRG.. No significant difference in any parameter was observed between the genotypes (t-Test, p>0.05).Error bars indicate s.e.m.

The following figure supplement is available for figure 6:

**Figure supplement 1.** Actin cytoskeletal organization in DRG from Atat1$^{cKO}$ mice.

Atat1$^{cKO}$ mice. However, since we observed no alteration in proton gated currents, it is unlikely that the reduction of mechanosensitivity is due to the altered transport of STOML3 in Atat1$^{cKO}$ mice.

Despite the profound loss of cutaneous mechanosensation, we observed no deficits in proprioception in Atat1$^{cKO}$ mice. This is in contrast to Piezo2 knockout mice which display severe proprioceptive defects (*Woo et al., 2015*). The lack of proprioceptive phenotype in Atat1$^{cKO}$ mice does not stem from incomplete recombination by the Avil-Cre driver in proprioceptive neurons because we previously observed that full Atat1 knockout mice also display no alterations in proprioception (*Kalebic et al., 2013b*). Rather it may arise either from developmental compensation of a partial reduction in mechanosensitivity in these neurons, or from mechanistic differences in mechanotransduction in proprioceptors and cutaneous afferents. Similarly, we did not observe here, or in ubiquitous knockout mice (*Kalebic et al., 2013b*), any apparent change in other aspects of mechanosensation, such as whisker function and exploration, or in developmental consequences of reduced touch sensitivity such as nursing or growth. This may reflect a lack of sensitivity of the tests we performed, or again, developmental compensation arising from residual responsiveness. The use of more selective conditional Cre lines would circumvent this problem and also allow for further studies on the role of Atat1 in specific touch sensations such as those arising from C-low threshold mechanoreceptors or from visceral mechanical nociceptors, which were not considered here.

We found that acetylated microtubules localize to a prominent band under the membrane of sensory neuron cell bodies. Furthermore our AFM measurements showed that in the absence of Atat1 and acetylated α-tubulin, the stiffness of sensory neurons was significantly elevated across a range of indentation stimuli in Atat1$^{cKO}$ mice, and cells were more resistant to osmotically induced shrinkage. We also monitored the movement of microtubules during shrinking using live imaging, and again observed reduced compression of the microtubule network upon Atat1 deletion. Importantly this defect could be rescued by introduction of the α-tubulin acetylation mimic into neurons. Thus our data point towards a role for acetylated tubulin in setting neuronal elasticity, and imply that in its absence, cells are stiffer which ultimately changes the force distribution to the neuronal membrane, requiring more force to indent the membrane and activate mechanosensitive channels. This is consistent with the phenotype in Atat1$^{cKO}$ mice in that all mechanoreceptor subtypes, regardless of the type of mechanosensitive currents/channels, are less responsive in the absence of Atat1. It also suggests that a residual response may be evident across patch clamping, skin-nerve electrophysiology and behavioural experiments, in that at higher forces, neuronal membranes in Atat1$^{cKO}$ mice would eventually be displaced, leading to mechanosensitive ion channel activation. Whether this is the case could be directly addressed using new technology based upon pillar arrays that displace neurons in the nanometer range allowing for quantitative analysis of mechanoelectrical transduction (*Poole et al., 2014*). Similarly, another approach which has been used previously (*Ranade et al., 2014*; *Wetzel et al., 2007*), would be to determine whether any afferents lose their mechanosensitivity completely in the ex vivo skin nerve preparation, which we did not investigate here.

While our data shows that there is a correlation between reduced mechanical sensitivity and neuronal elasticity in the absence of Atat1, there are some caveats to this interpretation. For example, we performed AFM measurements at the neuronal soma whereas in vivo, mechanotransduction occurs at sensory neuron endings. In future work it would therefore be informative to assess cell stiffness along neurites which are much more sensitive to mechanical displacement than the cell body (*Hu and Lewin, 2006*; *Poole et al., 2014*). Moreover, we detected acetylated tubulin enriched under the membrane of axons which are not in themselves mechanosensitive, and while we show that acetylated tubulin is expressed at neuronal endings in the cornea, superresolution microscopy would be required to demonstrate conclusively that it is indeed submembrane localized. In this light

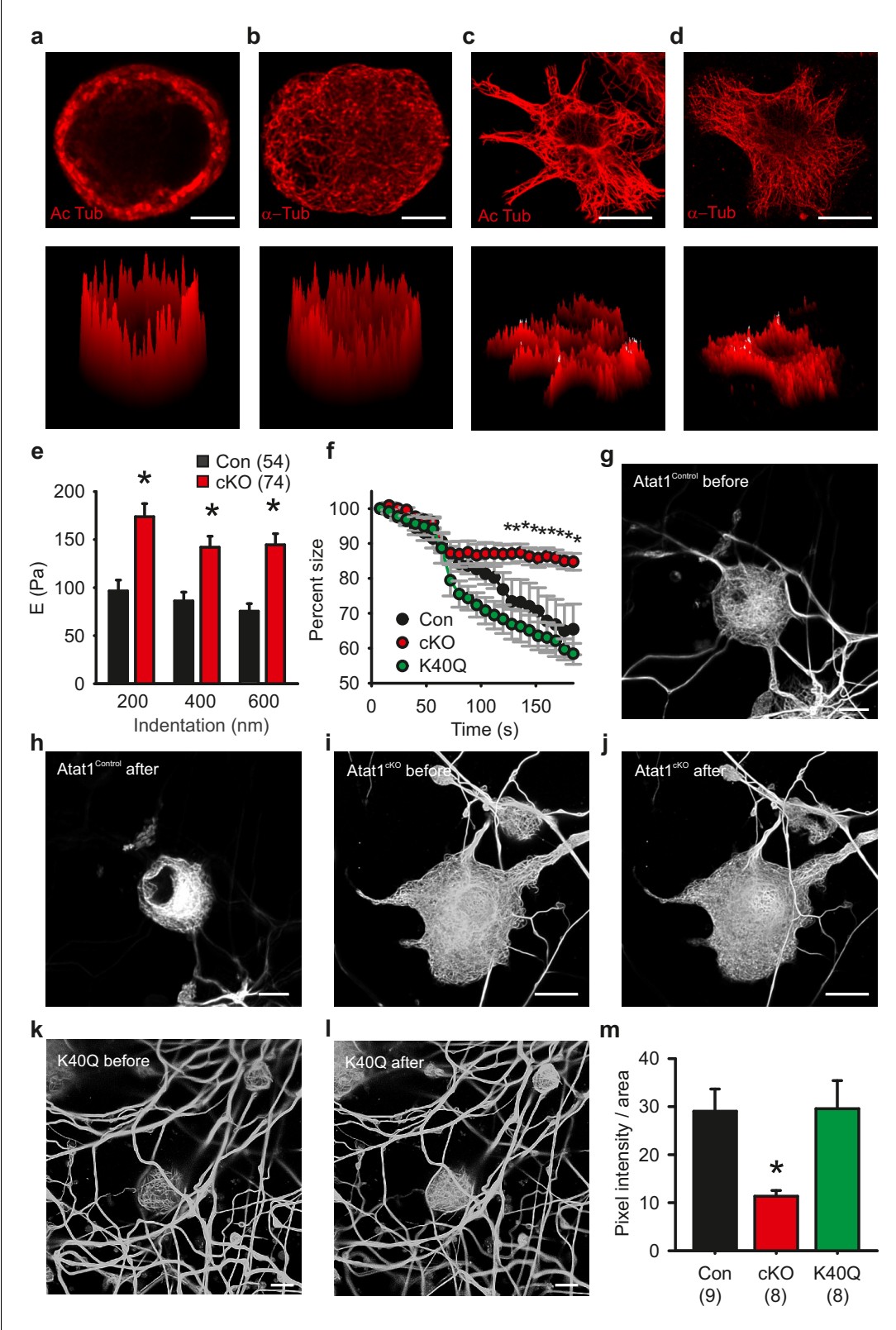

**Figure 7.** Microtubule organization in peripheral sensory neurons. (**a**) Anti-acetylated-α-tubulin staining of Atat1[Control] cultured DRG cells (corresponding surface plot below). Note the prominent sub-membrane localisation of acetylated tubulin (Scale bar 5 μm). (**b**) Anti α-tubulin staining of Atat1[Control] cultured DRG cells (Scale bar 5 μm) (**c**) Anti-acetylated-α-tubulin staining of serum-starved Atat1[Control] MEFs (Scale bar 20 μm). Note the even distribution of acetylated tubulin in this cell type. (**d**) Anti α-tubulin staining of serum-starved Atat1[Control] MEFs (Scale bar 20 μm). (**e**) Quantitative

*Figure 7 continued on next page*

*Figure 7 continued*

comparison of Young's modulus obtained by fitting force-indentation curves with the Hertz-Sneddon model at different indentations for cultured DRG taken from Atat1[Control] and Atat1[cKO] mice. A significantly higher pressure is required to indent the membranes of Atat1[cKO] neurons over Atat1[Control] cells (Mann-Whitney test, p<0.01). (f) Graph showing the relative shrinkage of axonal outgrowths from Atat1[Control] and Atat1[cKO] DRG loaded with calcein (2 μM) in response to a hyperosmotic shock over time. Deletion of Atat1 leads to a significant decrease in the percentage shrinking of Atat1[cKO] axons relative to control samples (ANOVA on ranks, multiple comparison Dunn's Method, p<0.05). (g–l) Live imaging of microtubules labelled with SiR Tubulin two and subjected to hyperosmotic shock (scale bars 10 μm) (g, h) Atat1[Control] DRG before and after hyperosmotic shock. (I, j) Atat1[cKO] DRG before and hyperosmotic shock. (k, l) Atat1[cKO] DRG transfected with the tubulin K40Q amino acid mimic before and hyperosmotic shock. (m) Bar-chart summarising osmotically induced microtubule compression in DRG neurons from Atat1[Control], Atat1[cKO], and Atat1[cKO] neurons transfected with tubulin-K40Q. There is significantly less compression in Atat1[cKO] than Atat1[Control] neurons, which is rescued by transfection of tubulin-K40Q (ANOVA on ranks, multiple comparison Dunn's Method, p<0.05). Error bars indicate s.e.m.

The following figure supplements are available for figure 7:

**Figure supplement 1.** Acetylated microtubule distribution in sensory neurons from Atat1[cKO] mice.

**Figure supplement 2.** Atomic Force Microscopy of DRG from Atat1[cKO] mice.

there are also other models of mechanosensation that would accord with our data. For example, microtubules have been implicated in sensory mechanotransduction in *Drosophila* larval dendritic arborisation neurons (*Zhang et al., 2015*), *C. elegans* touch receptor neurons (*Bounoutas et al., 2009*; *Fukushige et al., 1999*) and mammalian osmosensory neurons (*Prager-Khoutorsky et al., 2014*). Their role has mainly been explored from the perspective of an anchoring function that serves to tether mechanosensitive ion channels in the membrane. Further support for a tethered mechanotransduction complex comes from observations in sensory neurons where a physical tether has been described that may link mechanosensitive ion channels to the extracellular matrix (*Chiang et al., 2011*; *Hu et al., 2010*). It is possible that acetylated tubulin serves as a coordination point for these tethers in the mechanotransduction complex, and that in its absence misorganization of the microtubule network leads to reduced force transfer from the extracellular matrix to ion channels. Finally, another critical component of the complex, STOML3 has been demonstrated to change the force distribution in neuronal membrane via binding cholesterol. However, while a specific reduction of stiffness in sensory neuron membranes is observed in STOML3 knockout mice (*Qi et al., 2015*), cellular stiffness is increased upon deletion of Atat1, yet both STOML3 knockout mice and Atat1[cKO] mice exhibit deficits in mechanosensation. One explanation for this difference is that disruption of α-tubulin acetylation increases the overall neuronal stiffness, rendering sensory neurons more resistant to mechanical deformation, and thereby impeding the transmission of force to other mechanical transducing elements, including STOML3, the cytoskeleton, extracellular matrix and plasma membrane. In contrast, STOML3 distributes the force directly within the membrane to ion channels, and a stiffened membrane facilitates this process. It is also possible that membrane stiffness is set at an optimal range to distribute force among the membrane, cytoskeleton and extracellular matrix and thus tune the sensitivity of mechanosensitive ion channels.

In light of the increased stiffness of Atat1[cKO] sensory neurons we also examined the gross organization of the cytoskeleton in these cells. Similar to osmosensory neurons in the hypothalamus (*Prager-Khoutorsky et al., 2014*) we observed that microtubules formed a lattice like structure in DRG neurons; however this was not altered in the absence of Atat1. Similarly, we detected no difference in phalloidin staining in Atat1[cKO] mice suggesting that mechanosensory defects do not arise from interaction with the actin cytoskeleton. How α-tubulin acetylation sets

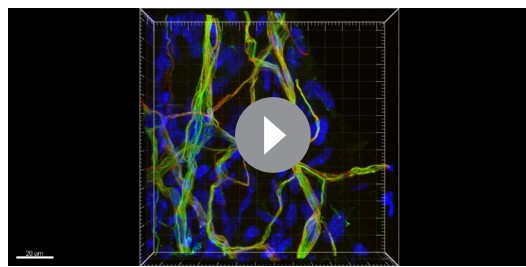

**Video 4.** Acetylated tubulin distribution in cornea. 3D rendering of a whole mount cornea preparation showing membrane bound SNAP labeling in red and acetylated tubulin in green.

cellular stiffness is therefore not immediately apparent, and a key question will be to investigate whether this occurs by directly altering the rigidity of individual microtubules or through interaction with other membrane and cytoskeletal elements. Intriguingly, it has recently been shown that another tubulin posttranslation modification, detyrosination, stiffens cardiomyocytes and promotes microtubule buckling in these cells (*Robison et al., 2016*). Desmin intermediate filaments were identified as a physical link between detyrosinated microtubules and the sarcomere, opening up the possibility that analogous mechanisms may function in sensory neurons, and that other microtubule posttranslational modifications may also influence sensory mechanotransduction. Further work using electron microscopy for example to examine the structure of microtubules and their modifications at the ultrastructural level, particularly under the membrane, may give insight into this process. What is clear however is that the mechanism appears to be broadly applicable since the lack of mechanosensitivity of Atat1[cKO] mice extends beyond phenotypes observed when other mechanotransduction complex proteins such as Piezo2 (*Ranade et al., 2014*) and STOML3 (*Wetzel et al., 2007*) are deleted. The convergence of multiple mechanisms onto a single posttranslational modification of α-tubulin, and its remarkable selectivity, indicates that microtubule acetylation represents a fundamental component of the mechanosensory apparatus, as well as an attractive target for novel therapeutic strategies to treat mechanical pain.

## Materials and methods

### Mouse lines
The generation and genotyping of all mouse lines used has been described previously (*Kalebic et al., 2013b*; *Yang et al., 2015*; *Zurborg et al., 2011*).

### Behavioural testing
The tape test (*Ranade et al., 2014*), cotton swab (*Ranade et al., 2014*), von Frey (*Kwan et al., 2006*), tail clip (*Ranade et al., 2014*), hot plate (*Kwan et al., 2006*), tail immersion test (*Elhabazi et al., 2014*), acetone drop (*Choi et al., 1994*) and rotarod assay (*Hamm et al., 1994*) respectively, were all performed as described previously. The grid test was performed by counting the number of steps versus slips over 2 min from a free roaming mouse placed atop a metal grid.

### Electrophysiological recordings
The skin nerve preparation was used essentially as previously described (*Wetzel et al., 2007*). Mice were sacrificed using CO2 inhalation, and the saphenous nerve together with the skin of the hind limb was dissected free and placed in an organ bath. The chamber was perfused with a synthetic interstitial fluid (SIF buffer) consisting of (in mM): NaCl, 123; KCl, 3.5; MgSO4, 0.7; NaH2PO4, 1.7; CaCl2, 2.0; sodium gluconate, 9.5; glucose, 5.5; sucrose, 7.5; and HEPES, 10 at a pH of 7.4.) The skin was placed with the corium side up, and the nerve was placed in an adjacent chamber for fiber teasing and single-unit recording. Single units were isolated with a mechanical search stimulus applied with a glass rod and classified by conduction velocity, von Frey hair thresholds and adaptation properties to suprathreshold stimuli. A computer-controlled nanomotor (Kleindiek Nanotechnik) was used to apply mechanical ramp-and-hold stimuli of known amplitude and velocity. Standardized displacement stimuli of 2 s or 10 s duration were applied to the receptive field at regular intervals (interstimulus period, 30 s). The probe was a stainless steel metal rod with a flat circular contact area of 0.8 mm. For thermal sensitivity, C fibers were tested with a thermal contact stimulator based on the Peltier principle (Thermal Stimulator, Yale University School of Medicine). Standard ramp-and hold heating stimulus (ramp 3°C per s, 32°C to 56°C, hold 5 s) was applied to the receptive field. The signal driving the movement of the linear motor and raw electrophysiological data were collected with a Powerlab 4.0 system and Labchart 7.1 software (AD instruments), Spikes were discriminated off-line with the spike histogram extension of the software.

### Patch clamping
DRG neurons were collected from mice and dissociated as described (*Wetzel et al., 2007*). Transfections were carried out using the Nucleofector system (Lonza AG) in 20 μl of Mouse Neuron Nucleofector solution from the SCN nucleofector kit (Lonza AG) and a total 4–5 μg of plasmid DNA at

room temperature using the preinstalled program SCN Basic Neuro program 6. After electroporation, the cell suspension was transferred to 500 µl of RPMI 1640 medium (Gibco) for 10 min at 37°C. This suspension, supplemented with 10% horse serum, was used to plate the cells onto glass coverslips for recording. The RPMI medium supplemented with 100 ng/ml nerve growth factor (NGF), 50 ng/ml BDNF was replaced with the standard DRG medium 3–4 hr later. Electrophysiology experiments began 12 hr after plating and only transfected cells (as determined by fluorescence) were selected for patch clamping. As a negative control, cells were transfected with an eGFP plasmid alone.

Whole-cell recordings from isolated DRG neurons were made as previously described (*Hu and Lewin, 2006*). Recordings were made from DRG neurons using fire-polished glass electrodes with a resistance of 3–7 MΩ. Extracellular solution contained (mM): NaCl 140, MgCl2 1, CaCl2 2, KCl 4, glucose four and HEPES 10 (pH 7.4), and electrodes were filled with a solution containing (mM): KCl 130, NaCl 10, MgCl2 1, EGTA one and HEPES 10 (pH 7.3). Cells were perfused with drug-containing solutions by moving an array of outlets in front of the patched cells (WAS02; Ditel, Prague). Observations were made with an Observer A1 inverted microscope (Zeiss) equipped with a CCD camera and the imaging software AxioVision. Membrane current and voltage were amplified and acquired using EPC-10 amplifier sampled at 40 kHz; acquired traces were analyzed using Patchmaster and Fitmaster software (HEKA). Pipette and membrane capacitance were compensated using the auto function of Pulse. For most of the experiments, to minimize the voltage error, 70% of the series resistance was compensated and the membrane voltage was held at −60 mV with the voltage-clamp circuit. After establishing whole-cell configuration, voltage-gated currents were measured using a standard series of voltage commands. Briefly, the neurons were pre-pulsed to −120 mV for 150 ms and depolarized from −65 to +55 mV in increments of 5 mV (40 ms test pulse duration). Next the amplifier was switched to current-clamp mode and current injection was used to evoke action potentials. If the membrane capacitance and resistance changed more than 20% after the mechanical stimulus, the cell was regarded as membrane damaged and the data discarded. Mechanical stimuli were applied using a heat-polished glass pipette (tip diameter 3–5 µm), driven by a MM3A Micromanipulator system (Kleindiek), and positioned at an angle of 45 degrees to the surface of the dish. For soma indentation, the probe was typically positioned so that a 1.4 µm movement did not visibly contact the cell but that a 2.8 µm stimulus produced an observable membrane deflection under the microscope. So a 2.8 µm probe movement was defined as a 1.4 µm mechanical stimulation, and a series of mechanical steps in 0.7 µm increment were applied at 5 s intervals. The displacement threshold was determined as mechanical stimulus that elicited a current ≥20 pA. For analysis of the kinetic properties of mechanically activated current, traces were fit with single exponential functions using the Fitmaster software (HEKA).

## Calcium imaging

Calcium imaging was performed as previously described (*Chen et al., 2014*). Fluorescence microscopy was done on an Observer A1 inverted microscope (Zeiss, Germany) using a 25 × 0.8 numerical aperture water immersion objective and a 175W Xenon lamp as a light source. Before imaging, DRG neurons were incubated with 4 mM Fura-2 at 37°C for 40 min and washed with extracellular solution at 37°C for another 30 min. Excitation light was passed either through a 340 BP 30 filter or a 387 BP 16 filter. Two filters were switched by an ultra-high speed wavelength switcher Lambda DG-4 (Sutter, Novato, CA). Emissions elicited from both excitation wavelengths were passed through a 510 BP 90 filter and collected by a charge-coupled device camera (Zeiss). Different solutions were applied by multi barrel perfusion system (WAS02, DITEL, Prague). AxioVision software (Zeiss) was used to record image data. After background ($B_{340}$, $B_{380}$) subtraction in each channel ($F_{340}$, $F_{380}$), the ratio (R) of fluorescence elicited by two excitation light was calculated: $R=(F_{340} - B_{340}) / (F_{380} - B_{380})$. Data was analyzed in AxioVision, Matlab (MathWorks, Natick, Massachusetts), and GraphPad prism (GraphPad Software Inc., San Diego, CA).

## DRG neurons culture and transfection

DRG neurons from adult male mice were prepared as previously described (*Hu and Lewin, 2006*). Briefly, DRGs were dissected and collected in a 1.0 ml tube of phosphate-buffered saline (PBS) on ice. Ganglia were cleaned, enzymatically treated and mechanically dispersed. Before plating on

poly-l-lysine–laminin coated coverslips, cells were transfected using the Nucleofector system (Lonza AG). In brief, neurons were suspended in 20 μl of Mouse Neuron Nucleofector solution from SCN nucleofector kit (Lonza AG) and a total 4–5 μg of plasmid DNA at room temperature. The mixture was transferred to a cuvette and electroporated with the preinstalled program SCN Basic Neuro program 6. After electroporation, the cell suspension was transferred to 500 μl of RPMI 1640 medium (Gibco) for 10 min at 37°C. This suspension, supplemented with 10% horse serum, was used to plate the cells onto glass coverslips for recording. The RPMI medium supplemented with 100 ng/ml nerve growth factor (NGF), 50 ng/ml BDNF was replaced with the standard DRG medium 3–4 hr later. Electrophysiology experiments began 12 hr after plating.

## Immunofluorescence and staining

For microtubule staining in DRG cultures, cells were washed once with PBS, and then fixed for 15 min in cytoskeleton buffer (CB) pH 6.3 containing 3% paraformaldehyde, 0.25% triton and 0.2% glutaraldehyde at room temperature. Cells were then washed three times with PBST (0.3% triton). Samples were then subsequently blocked with 5% normal goat serum (NGS) in PBS for 1 hr at room temperature. Cells were then placed overnight at 4°C with primary anti α-tubulin (1:1000) (Sigma-Aldrich, T9026) or anti-acetylated-α-tubulin (1:1000) (Sigma-Aldrich, T7451) in PBS. Cells were then washed with PBS and incubated for 1 hr with fluorescently labelled secondary antibodies (1:1000) (Alexa Fluor 546 Lifetechnologies) for 1 hr at room temperature. All images were acquired using a 40 X objective on a Leica SP5 confocal microscope. Processing of images and generation of surface plots were performed using ImageJ. Images were deconvoluted using Huygens software.

Actin filaments were stained with phalloidin using the protocol outlined (*Cramer and Mitchison, 1995*). Briefly, actin filaments in DRG primary cultures were stained with Alexa488-phalloidin at 0.5 μg/ml (Lifetechnologies).Cells were fixed with fresh 4% PFA (EM grade, TAAB) in cytoskeleton buffer (10 mM MES, 138 mM KCl, 3 mM MgCl, 2 mM EGTA) freshly added supplemented of 0.3 M sucrose, permeabilized in 0.25% Triton-X-100 (Sigma-Aldrich), and blocked in 2% BSA (Sigma-Aldrich).

Immunostaining of saphenous nerves was performed on paraffin sections after fixation with PFA. Following rehydration, antigen retrieval was performed with 10 mM sodium citrate (pH 6) at boiling temperature for 10 min. Subsequently, sections were permeabilized (0.3% Triton X-100), blocked (5% goat serum) and stained with anti-acetylated-α-tubulin (Sigma-Aldrich, T7451) and anti-myelin basic protein (Chemicon, MAB386).

For cornea staining, the eyes were removed and fixed for 1 hr in 4% PFA at room temperature. The cornea was then dissected and permeabilized with PBS-Triton 0.03% for 30 min. Following this, the cornea was immersed in PBS-Triton 0.03% containing 1 μM SNAP surface 546 (New England Biolabs) for 30 min. The samples were then washed with PBS-Triton 0.03% for 20 min and subsequently blocked with 5% normal goat serum in PBS-Triton 0.03% for 30 min. The tissue was then stained with anti-acetylated-α-tubulin (1:500) overnight. Samples were then washed with PBS and a secondary antibody (Alexa Fluor 488 Lifetechnologies) was added for 5 hr. The samples were again washed with PBS and stained with DAPI 10 min. The cornea was then mounted on glass with 100% glycerol and imaged.

## Axon outgrowth assay

For whole mount axon outgrowth assays, individual DRG were extracted from mice and grown in Matrigel (Corning) for seven days. Preparations were fixed with 4% PFA for 5 min and labelled with the primary antibody PGP9.5 (1:200) overnight at 4°C. The samples were then labelled with secondary antibodies (1:1000) Alexa Fluor 546 Lifetechnologies) for 1 hr at room temperature. All images were acquired using a Leica LMD 7000.

## SNAP-tag skin labelling

SNAP-tag labelling was carried out by intradermal injection of the finger in anaesthetized mice of 2 μM BG TMRstar substrate as described previously (*Yang et al., 2015*). After five hours the animals were sacrificed and the samples were mounted in 80% glycerol for imaging.

## Electron microscopy of saphenous nerve

Saphenous nerves were postfixed for 24 hr with 4% PFA, 2.5% glutaraldehyde (TAAB) in 0.1 M phosphate buffer at 4C. Then incubated for 2 hr with 1% osmium tetraoxide with 1.5% potassium ferrocyanide, then dehydrated in ethanol and embedded in Epon for ultrathin sectioning and TEM imaging.

## Spinal cord staining of the dorsal horn

Spinal cord was extracted and fixed for 3 hr in 4% PFA at 4°C. Samples were embedded in 2% agarose and 100 µM sections were cut. The sections were treated with 50% ethanol 30 min and subsequently incubated with a 5% serum blocking solution of 0.3% Triton-X in PBS for 1 hr. Samples were then incubated with the primary antibodies overnight at 4°C NF200 1:200 (Sigma Aldrich, N0142), CGRP 1:500 (Rockland, 200–301-DI5) and IB4 1:100 (Invitrogen, I21414). Sections were washed with PBS, and incubated for 1 hr with the secondary antibody 1:1000 (Alexa Fluor 488 and 546 Life technologies) in 5% serum blocking solution. The samples were then mounted with prolong gold (Invitrogen, P36930) for imaging.

## Microfluidics

DRG neurons were suspended in 1:1 Matrigel in 10% FBS DMEM and seeded onto a two-chamber microfluidic chip (Xona Microfluidics, SD150). Axons were allowed to grow across the microchannels connecting the two chambers for 3–5 days. On the day of the experiment, media in both the cell body and axon chambers was replaced with media with no serum for 3 hr. 1 µM mono-biotinylated NGF purified in house from eukaryotic cells was coupled with 1 µM streptavidin conjugated quantum dots 655 (Life Technologies, Italy) for 30 min on ice, then diluted to 5 nM in imaging buffer (as above) and then used to replace the media in the axon chamber. A 25% vol difference was kept between the cell body and the axon chamber to avoid backflow from the axon to the cell body chamber. After 1 hr incubation at 37°C in 5% $CO_2$, retrograde transport of NGF-Qdot655 containing endosomes was imaged using a confocal Ultraview Vox (Perkin Elmer) equipped with a 5% $CO_2$ humidified chamber at 37°C. 100 s time lapses were recorded using 300 ms exposure time. Images were analyzed with Imaris software using the particle tracking function and autoregressive motion track generation setting.

## FRAP analysis of DRG

DRG were transfected with Piezo2-GFP using a nucleofector device (Amaxa) as described above. After two days FRAP analysis was then carried out on a LEICA SP5 confocal microscope. Cells were imaged for 10 frames pre-bleach before being bleached for a further 10 frames (70% laser power), following this the cells were imaged once every minute for 15 min to measure recovery. Data was analysed and graphs generated using Sigma-Plot.

## Superresolution microscopy

The cells were washed once with 3 ml of warm PBS and then fixed and permeabilized for 2 min in cytoskeleton buffer containing 0.3% Glutaraldehyde and 0.25% Triton X-100. Following this, the cells were fixed for 10 min in cytoskeleton buffer containing 2% Glutaraldehyde and treated for 7 min with 2 ml of 0.1% Sodium Borohydride (NaBH4) in PBS. Cells were then washed 3 times for 10 min in PBS. The cells were incubated with primary antibody for 30 min (mouse anti α-tubulin, Neomarker, 1:500) in PBS +2% BSA After washing 3 times for 10 min with PBS, the cells were transferred to the secondary antibody (goat anti mouse Alexa 647, 1:500, Molecular Probes A21236) at room temperature for 30 min. The cells were then washed three times with PBS for 10 min and then mounted for STORM imaging. At the time of imaging cells were overlaid with STORM blinking buffer: 50 mM Tris pH 8.0, 10 mM NaCl, 10% Glucose, 100 U/ml Glucose Oxidase (Sigma-Aldrich), 40 µg/ml Catalase (Sigma-Aldrich).

## Superresolution image analysis

The analysis of microtubule (MT) network morphology was done using the open source software CellProfiler (*Carpenter et al., 2006*). The MT signal was enhanced by a top-hat filter and then binarised with the same manual threshold for all images. Binary images were skeletonized using CellProfiler's 'skelPE' algorithm and the resulting skeleton was subjected to branchpoint detection. We

measured MT density by dividing the skeleton length with imaged cell area and we measured MT crossing density by dividing the number of branchpoints with skeleton length. Moreover, we measured the local angular distribution of the MTs in order to assess whether they run in parallel, or in a crossing manner (angular variance). To this end, we subjected each pixel to a rotating morphological filter using a linear structural element with a length of 11 pixels, and recorded at which angle we obtained a maximum response. We computed the response for angles from 0 to 170 degrees at steps of 10 degrees since there is no information on MT polarity. Next we measured the local circular variance of the MT orientations in a sliding window with a diameter of 51 pixels, using angle doubling as it is commonly done for axial data. The circular variance has a value one if the MTs in a given region are completely parallel and has smaller values (down to 0) if the MTs are oriented in various directions. Finally, we computed the average circular variance of all MT pixels in a given cell. If this value were close to one it would mean that locally, on a length scale of 51 pixels, the MTs are parallel in most of the cell.

## Atomic force microscopy (AFM)

Force spectroscopy measurements were performed in a similar manner to that described previously (*Qi et al., 2015*). Force spectroscopy measurements were performed by using a NanoWizard AFM (JPK Instruments, Berlin, Germany) equipped with a fluid chamber (Biocell; JPK) for live cell analysis and an inverted optical microscope (Axiovert 200; Zeiss) for sample observation.

DRG cells were seeded on glass coverslips previously coated with a first layer of polylysine (500 µg/ml for about 1 hr room temperature) and a second layer of laminin (20 µg/ml for about 1 hr at 37°C). The cells were then cultured for at least 15 hr before measurements. Then, the sample was inserted into the fluid chamber immersed in culture medium and measurements were carried out at room temperature. The status of cells was constantly monitored by optical microscope.

Indenters for probing cell elasticity were prepared by mounting silica microspheres of 4.5 µm nominal diameter (Bangs Laboratories Inc.) to tipless V-shaped silicon nitride cantilevers having nominal spring constants of 0.32 N/m or 0.08 N/m (NanoWorld, Innovative Technologies) by using UV sensitive glue (Loxeal UV Glue). Silica beads were picked under microscopy control. Before measurements the spring constant of the cantilevers was calibrated by using the thermal noise method.

By using the optical microscope the bead-mounted cantilever was brought over the soma of single DRG and pressed down to indent the cell. The motion of the z-piezo and the force were recorded. On each cell eight-about ten force-distance curves were acquired with a force load of 500 pN and at a rate of 5 µm sec-1 in closed loop feed-back mode.

Cell elastic properties were assessed by evaluating the Young's modulus (E) of the cell. This value was obtained by analysing the approaching part of the recorded F-D curves using the JPK DP software. The software converts the approaching curve into force-indentation curves by subtracting the cantilever bending from the signal height to calculate indentation. Afterwards force-indentation curves were fitted by Hertz-Sneddon model for a spherical indenter according to this equation:

$$F = \frac{E}{1-\nu^2}\left[\frac{a^2 + R_s^2}{2}\ln\frac{R_s + a}{R_s - a} - aR_s\right]$$

$$\delta = \frac{a}{2}\ln\frac{R_s + a}{R_s - a}$$

Here, $\delta$ is the indentation depth, a is the contact radius of the indenter, R is the silica bead radius, $\nu$ is the sample's Poisson ratio (set to 0.5 for cell) (*Rotsch et al., 1999*) and E is Young's modulus. Fitting was performed at different indentations 200, 400 and 600 nm (see SI for examples of fitting curves obtained).

## Osmotic shrinking assays

Cultured DRG were loaded with 500 nM SiR Tubulin2 for 1 hr at 37°C and/or 2 µM calcein dye (Invitrogen C3100MP) for 30 min in DRG at 37°C. The cells were then transferred to imaging buffer (10 mM Hepes pH 7.4, 140 mM NaCl, 4 mM KCl, 2 mM CaCl2, 1 mM MgCl2, 5 mM D-glucose) at

320mOsm. Following a 5 min acclimatization period the cells were subjected to a 440mOsm (osmolarity adjusted with mannitol) hyperosmotic shock for 3 min. All imaging was carried out using a Leica SP5 resonant scanner. To increase the permeability and fluorogenicity of the previously described far-red microtubule probe SiR-tubulin2 (*Lukinavičius et al., 2014*) we exchanged the docetaxel in SiR-tubulin with the more hydrophobic cabazitaxel. The molecule was synthesized using the same procedures as described for SiR-tubulin. A detailed characterization of SiR-tubulin2 will be published elsewhere.

## Acknowledgements

We thank Maria Kamber and Emerald Perlas of EMBL Monterotondo Mouse and Histology facilities for technical support of our work and Violetta Paribeni for mouse husbandry. Funded by EMBL and DFG.

## Additional information

### Funding

| Funder | Author |
| --- | --- |
| Deutsche Forschungsgemeinschaft | Paul A Heppenstall |

The funders had no role in study design, data collection and interpretation, or the decision to submit the work for publication.

### Author contributions

SJM, JH, PAH, Conception and design, Acquisition of data, Analysis and interpretation of data, Drafting or revising the article; YQ, LI, DG, NK, LC, CT, CP, GB, KS, CMF, AA, FF, MS, UM, ML, JR, YS, Acquisition of data, Analysis and interpretation of data; LA, Acquisition of data, Analysis and interpretation of data, Drafting or revising the article; LR, ADN, LB, KJ, Contributed unpublished essential data or reagents

### Author ORCIDs

Jonas Ries, http://orcid.org/0000-0002-6640-9250
Yannick Schwab, http://orcid.org/0000-0001-8027-1836
Paul A Heppenstall, http://orcid.org/0000-0001-7954-5166

### Ethics

Animal experimentation: This study was performed in strict accordance with the recommendations in the Guide for the Care and Use of Laboratory Animals of the National Institutes of Health. Mice were maintained at the EMBL Mouse Biology Unit, Monterotondo, Italy, in accordance with Italian legislation (Art. 9, 27. Jan 1992, no 116) under license from the Italian Ministry of Health.

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
