## [Decision Letter]

Thank you for submitting your article "Acetylated tubulin is essential for touch sensation in mice" for consideration by *eLife*. Your article has been favorably evaluated by Anna Akhmanova as the Senior Editor and three reviewers, one of whom is a member of our Board of Reviewing Editors. The reviewers have opted to remain anonymous.

The reviewers have discussed the reviews with one another and the Reviewing Editor has drafted this decision to help you prepare a revised submission.

Summary:

In this paper Morley and colleagues have generated a conditional mutation of the α-tubilin acetylase gene Atat1 in sensory neurons using an advillin-Cre crossed with a flossed allele of Atat1. The authors report profound deficits in the behavior of the animals when confronted with a variety of mechanical stimuli but no impairments in their responses to thermal stimuli. Analysis of the physiological responses of identified primary afferents innervating the hairy skin indicate that loss of Atat1 leads to profound loss of mechanosensitivity amongst both mechanoreceptors and nociceptors. The authors report that loss of mechanosensitivity is associated with loss of mechanosensitive currents evoked by poking isolated sensory neurons. Importantly, the authors show that the loss of mechanosensitive current can be rescued by reintroducing Atat1 into the neurons or by a mutation in α-tubilin that mimics the effects of the acetylation. The authors have carried out an extensive series of control experiments in an attempt to rule out indirect effects of Atat1 deletion during development that might impact on the phenotype. Finally, the authors conclude that acetylation of α-tubilin may be necessary to stiffen the membrane which enhances the ability of force to activate mechanosensitive channels. The paper contains novel and interesting data that will have a significant impact on the field and will be of broad interest to many neuroscientists working on sensory mechanisms. The key conclusion that the authors make that it is membrane stiffness that "explains" the effect of the Atat1 gene deletion is however overstated and based on a selective review of the literature. The authors should consider alternative explanations and discuss more critically the interpretation of their experiments.

Essential revisions:

All reviewers mention that the paper contains novel and interesting data that will have a significant impact on the field and will be of broad interest to many neuroscientists working on sensory mechanisms. However, there are a number of major concerns, which preclude publication in its present form. Some concerns mention a lack of related references and a discussion of alternative models.

The authors are invited to submit a revised version of the manuscript, in which they fully address all points listed below (points 1-3 require additional experimental data).

1) Rotarod and Grid test (Figure 1): the authors conclude that Atat1 KO mice are normal in motor coordination, since no significant differences were observed between the genotypes. This may be true, however the number of animals per experiment was very low with six and seven individuals per genotype, respectively. In contrast to the acetone test, which is also based on six mice each, rotarod and grid test show a tendency and large error bars (e.g. KOs in rotarod). N-values in rotarod and grid test, need to be increased to support the statements on proprioception.

2) The authors used heterozygous KO mice as controls throughout the study. They may have done pilot experiments to prove that (i) heterozygotes perform similar as wildtypes and (ii) Cre expression per se has no effect. This needs to be stated in the manuscript or to be added as supplemental controls to support the conclusions.

3) Please add data on the efficiency of Atat gene deletion in the conditional mice.

Is the phenotype of reduced sensory neuron mechanosensitivity identical in the Atat1 full knockout? Cre drivers are never 100% efficient in recombining with floxed alleles and the efficiency for different floxed loci are not identical. I did not see any data on the efficiency of Atat1 gene deletion in the cKO mice. qPCR, southern blotting etc. are not shown. The easiest way to be sure here is to show that the phenotype in the full knockout is at least as severe as in the cKO. Have these experiments been done?

4) The key conclusion that the authors make that it is membrane stiffness that "explains" the effect of the Atat1 gene deletion is however overstated and based on a selective review of the literature. The authors should consider alternative explanations and discuss more critically the interpretation of their experiments.

5) Please cite and discuss the following literature based on the comments below. Please consider and discuss alternative explanations and models.

A) The conclusion that changes in membrane stiffness underlies the effect of Atat1 gene deletion is based on data from AFM measurements on the cell body in culture. This is not the place where transduction happens in vivo and more importantly is completely different in morphology to the sensory endings of any sensory fiber type in the skin (tiny elongated tubes compared to a huge compressible balloon). This is fine when measuring the activation of mechanosensitive currents at the cell soma but there are several papers showing that displacement thresholds are much smaller to activate such currents in sensory neurites which are much more similar to sensory ending in vivo (e.g. PMID: 21725315 and Hu and Lewin 2006 quoted, but not in this context).

B) Moreover, the authors suggest that stimuli in a range up to 200 nm indentation may be selective for indentation of the membrane whereas up to 600 nm may impinge on the cytoskeleton. However, in a recent study Poole and colleagues showed that membrane displacements of much less than 50 nm are capable of maximally activating mechanosensitive currents in mechanoreceptors (PMID:24662763, not quoted). Indeed, their finding that membrane stiffness increases after deletion of Atat1 is to this reviewer counterintuitive as increased stiffness of the membrane would be expected to decrease the activation threshold for mechanosensitive channels to force. However, this expectation is highly dependent on where channels are in relation to other mechanical elements in the system. Indeed, the authors totally ignore models, which are also based on good experimental evidence in sensory neurons (see PMID: 20075867; 21725315) that physical tethers that link directly or indirectly channels to the matrix are critical for the sensitivity of mechanosensitive channels to force. It appears to me more likely that the organization or mis-organization of the microtubule network in the vicinity of such complexes that include matrix interaction may better explain the loss of mechanoreceptor sensitivity in their mutant mice. In any case ignoring the literature that such interactions are critically important oversimplifies and may mislead the reader. Again, the change in stiffness of the membrane correlates with the effects on mechanosensitivity but does not necessarily explain them.

6) One additional point is that Hu and colleagues have claimed that membrane softening is associated with loss of STOML3 (PMID 26443885) but here the authors observe membrane stiffening in the absence of Atat1. However, both genetic manipulations lead to loss of mechanosensitive currents. This is not so easily reconciled with the authors favoured interpretation and should be explicitly addressed in the Discussion. (Posted 27th Sep 2016)

7) Introduction, third paragraph: "well conserved in ciliated organisms" – organisms are not ciliated they have cells with cilia. Reword please.

8) Subsection “Atat1^cKO^ mice display reduced mechanosensitivity across all mechanoreceptor subtypes innervating the skin”, second paragraph: Rugiero et al. 2010 showed that the RA mechanosensitive current is velocity dependent not that mechanoreceptor firing frequencies are highly velocity dependent an issue addressed directly in PMID 18815344 for instance. Please replace with a more appropriate reference.

9) Subsection “Morphological analysis of the peripheral nervous system in Atat1^cKO^ mice”: "neurotrophin transport" is misleading; it is neurotrophin/receptor transport as such experiments are generally thought to measure transport of NGF internalized with its receptor. It is odd here that the authors fail to mention their own work indicating that STOML3 another critical mediator of mechanotransduction is transported in vesicles associated with microtubules (PMID 22773952). Indeed, this paper shows that then molecular characteristics of such vesicles are distinct from classical signaling endosomes so a change in STOML3 transport could conceivably contribute to the phenotype they see. This is worth mentioning and emphasizes that the interpretation that membrane stiffness is everything is being over sold at the expense of other possibilities.

10) Subsection “Genetically mimicking α-tubulin acetylation restores mechanosensitivity of Atat1^cKO^ sensory neurons”. The authors repeatedly emphasize the requirement of Atat1 for mechanosensitive currents. However, it has recently been shown that lack of mechanosensitive current activation to poking is not associated with complete loss of mechanosensitive currents (PMID:24662763) as membrane displacement experiments demonstrate that loss of mechanosensitive currents in STOML3 mutant neurons is associated not with loss but with increased displacement thresholds for activating native emchanosensitive neurons (PMID:24662763). The authors should modulate their language accordingly.

11) This raises another issue that the authors did not address experimentally, which at least deserves a mention. Namely: were any of their recorded afferents in the ex vivo skin nerve preparation completely silent to mechanical stimulation? If the authors have not measured this it should be at least mentioned that this is an unknown.

12) Subsection “Increased rigidity of DRG neurons from Atat1^cKO^ mice”, first paragraph: rather arbitrary statements about the indentations that impinge on membrane or cytoskeleton. The plasma membrane is less than 10 nm thick so I fail to see why 200nm should primarily perturb the plasma membrane only (see again PMID 24662763).

13) Discussion, sixth paragraph: how can loss of mechanosensitivity in nociceptors (which is in any case no by no means complete) lead to noxious stimuli being perceived as innocuous? This speculation should be removed.

14) The Discussion should include more alternative models and not focus exclusively on membrane stiffness as the only explanation. Several important parts of the literature have been completely ignored.

15) I cannot see whether the experiments measuring mechanosensitive currents in DRG neurons was done blind to the genotype and/or transfection. Please mention in the manuscript.

16) The last part of the Results section, mainly data presented in Figure 7, is less convincing. Figure 7 show acetylated tubulin in MEFs. I was very surprised by the image presented since similar experiments performed in other labs (using the same antibody) tend to only reveal the primary cilium. Why is the entire microtubular network visible here? Has the microtubule network been stabilized in some way (as was the case in another paper from the Heppenstall lab)? If so, it is not mentioned in the figure legend and/or Materials and methods section. Please add the missing information. This possible problem has no impact on the overall meaning of the figure (no sub-membrane localization of acetylated tubulin in MEFs) but it does need to be explained/corrected.

17) The presence of a sub-cortical acetylated tubulin band in the cell body of DRG neurons (Figure 7) is very convincing. In contrast, data for saphenous nerve axons and sensory neuron axons are less convincing. To fully conclude, super-resolution imaging of axons will be needed, although I know this is hard to perform. To circumvent this requirement, I suggest that the authors use less affirmative phrasing in the Results/figure sections (subsection “Acetylated tubulin localizes to a prominent submembrane band in peripheral sensory neurons” and Figure 7 legend), that they provide higher-magnification images, or that they confine these results to the Supplemental data.

18) The data relating to hyperosmotic experiments, presented in Figure 7, are not adequate; a before and after picture is required (as shown in the associated supplementary figure). In addition, the quantification indicated that 9 cells were analyzed in control conditions, but the cells shown in Figure 7 and in the supplementary data appear to be the same one. Could you provide another picture? Overall I think that the data relating to hyperosmotic shock, which is currently presented in the supplemental figure, should be included in Figure 7 in the main text.

19) The Discussion is very interesting. I would strongly suggest that the authors tone down the sentence "We found that acetylated microtubules localize to a prominent band under the membrane of sensory neuron cell bodies and axons". For me, the presence of such a band in axons is not clearly demonstrated in the manuscript, even though I agree that it is likely to exist.

20) A recent publication established direct links between tubulin tyrosination and cellular stiffness in myocytes (Robison et al. 2016 Science). These results should be mentioned in the Discussion, and possible direct links between tubulin acetylation and detyrosinated microtubules should be discussed.

---

## [Author Response]

*Essential revisions:*

*All reviewers mention that the paper contains novel and interesting data that will have a significant impact on the field and will be of broad interest to many neuroscientists working on sensory mechanisms. However, there are a number of major concerns, which preclude publication in its present form. Some concerns mention a lack of related references and a discussion of alternative models.*

*The authors are invited to submit a revised version of the manuscript, in which they fully address all points listed below (points 1-3 require additional experimental data).*

*1) Rotarod and Grid test (Figure 1): the authors conclude that Atat1 KO mice are normal in motor coordination, since no significant differences were observed between the genotypes. This may be true, however the number of animals per experiment was very low with six and seven individuals per genotype, respectively. In contrast to the acetone test, which is also based on six mice each, rotarod and grid test show a tendency and large error bars (e.g. KOs in rotarod). N-values in rotarod and grid test, need to be increased to support the statements on proprioception.*

We have performed additional experiments to increase N numbers. This data has been added to Figure 1.

*2) The authors used heterozygous KO mice as controls throughout the study. They may have done pilot experiments to prove that (i) heterozygotes perform similar as wildtypes and (ii) Cre expression per se has no effect. This needs to be stated in the manuscript or to be added as supplemental controls to support the conclusions.*

We have stated in the manuscript that heterozygote control mice perform similarly to wildtypes. Data is also shown in Figure 1—figure supplement 1 supporting this statement.

*3) Please add data on the efficiency of Atat gene deletion in the conditional mice.*

*Is the phenotype of reduced sensory neuron mechanosensitivity identical in the Atat1 full knockout? Cre drivers are never 100% efficient in recombining with floxed alleles and the efficiency for different floxed loci are not identical. I did not see any data on the efficiency of Atat1 gene deletion in the cKO mice. qPCR, southern blotting etc. are not shown. The easiest way to be sure here is to show that the phenotype in the full knockout is at least as severe as in the cKO. Have these experiments been done?*

We have added data to Figure 1—figure supplement 1 showing that the phenotype in the full knockout is as severe as the cKO.

*4) The key conclusion that the authors make that it is membrane stiffness that "explains" the effect of the Atat1 gene deletion is however overstated and based on a selective review of the literature. The authors should consider alternative explanations and discuss more critically the interpretation of their experiments.*

We have expanded the Discussion to include alternative mechanisms and be more critical of our own data.

*5) Please cite and discuss the following literature based on the comments below. Please consider and discuss alternative explanations and models.*

*A) The conclusion that changes in membrane stiffness underlies the effect of Atat1 gene deletion is based on data from AFM measurements on the cell body in culture. This is not the place where transduction happens in vivo and more importantly is completely different in morphology to the sensory endings of any sensory fiber type in the skin (tiny elongated tubes compared to a huge compressible balloon). This is fine when measuring the activation of mechanosensitive currents at the cell soma but there are several papers showing that displacement thresholds are much smaller to activate such currents in sensory neurites which are much more similar to sensory ending in vivo (e.g. PMID: 21725315 and Hu and Lewin 2006 quoted, but not in this context).*

This point has been discussed.

*B) Moreover, the authors suggest that stimuli in a range up to 200 nm indentation may be selective for indentation of the membrane whereas up to 600 nm may impinge on the cytoskeleton. However, in a recent study Poole and colleagues showed that membrane displacements of much less than 50 nm are capable of maximally activating mechanosensitive currents in mechanoreceptors (PMID:24662763, not quoted). Indeed, their finding that membrane stiffness increases after deletion of Atat1 is to this reviewer counterintuitive as increased stiffness of the membrane would be expected to decrease the activation threshold for mechanosensitive channels to force. However, this expectation is highly dependent on where channels are in relation to other mechanical elements in the system. Indeed, the authors totally ignore models, which are also based on good experimental evidence in sensory neurons (see PMID: 20075867; 21725315) that physical tethers that link directly or indirectly channels to the matrix are critical for the sensitivity of mechanosensitive channels to force. It appears to me more likely that the organization or mis-organization of the microtubule network in the vicinity of such complexes that include matrix interaction may better explain the loss of mechanoreceptor sensitivity in their mutant mice. In any case ignoring the literature that such interactions are critically important oversimplifies and may mislead the reader. Again, the change in stiffness of the membrane correlates with the effects on mechanosensitivity but does not necessarily explain them.*

We have discussed more fully our data in light of other models of mechanosensation adding the suggested references.

*6) One additional point is that Hu and colleagues have claimed that membrane softening is associated with loss of STOML3 (PMID 26443885) but here the authors observe membrane stiffening in the absence of Atat1. However, both genetic manipulations lead to loss of mechanosensitive currents. This is not so easily reconciled with the authors favoured interpretation and should be explicitly addressed in the Discussion. (Posted 27th Sep 2016)*

We have discussed further membrane softening associated reductions in mechanosensitivity in light of our data.

*7) Introduction, third paragraph: "well conserved in ciliated organisms" – organisms are not ciliated they have cells with cilia. Reword please.*

We have corrected this.

*8) Subsection “Atat1^cKO^ mice display reduced mechanosensitivity across all mechanoreceptor subtypes innervating the skin”, second paragraph: Rugiero et al. 2010 showed that the RA mechanosensitive current is velocity dependent not that mechanoreceptor firing frequencies are highly velocity dependent an issue addressed directly in PMID 18815344 for instance. Please replace with a more appropriate reference.*

We have replaced with the correct reference.

*9) Subsection “Morphological analysis of the peripheral nervous system in Atat1^cKO^ mice”: "neurotrophin transport" is misleading; it is neurotrophin/receptor transport as such experiments are generally thought to measure transport of NGF internalized with its receptor. It is odd here that the authors fail to mention their own work indicating that STOML3 another critical mediator of mechanotransduction is transported in vesicles associated with microtubules (PMID 22773952). Indeed, this paper shows that then molecular characteristics of such vesicles are distinct from classical signaling endosomes so a change in STOML3 transport could conceivably contribute to the phenotype they see. This is worth mentioning and emphasizes that the interpretation that membrane stiffness is everything is being over sold at the expense of other possibilities.*

We have corrected neurotrophin transport to neurotrophin/receptor transport. We have also discussed our previous finding that STOML3 is transported in microtubule associated vesicles and that this could also underlie the phenotype.

*10) Subsection “Genetically mimicking α-tubulin acetylation restores mechanosensitivity of Atat1^cKO^ sensory neurons”. The authors repeatedly emphasize the requirement of Atat1 for mechanosensitive currents. However, it has recently been shown that lack of mechanosensitive current activation to poking is not associated with complete loss of mechanosensitive currents (PMID:24662763) as membrane displacement experiments demonstrate that loss of mechanosensitive currents in STOML3 mutant neurons is associated not with loss but with increased displacement thresholds for activating native emchanosensitive neurons (PMID:24662763). The authors should modulate their language accordingly.*

We have corrected statements where we over emphasize the requirement of Atat1 for mechanosensitive currents. We have also discussed the possibility that increased displacement thresholds could account for the phenotype.

*11) This raises another issue that the authors did not address experimentally, which at least deserves a mention. Namely: were any of their recorded afferents in the ex vivo skin nerve preparation completely silent to mechanical stimulation? If the authors have not measured this it should be at least mentioned that this is an unknown.*

This is an important point that we have added to the Discussion.

*12) Subsection “Increased rigidity of DRG neurons from Atat1^cKO^ mice”, first paragraph: rather arbitrary statements about the indentations that impinge on membrane or cytoskeleton. The plasma membrane is less than 10 nm thick so I fail to see why 200nm should primarily perturb the plasma membrane only (see again PMID 24662763).*

We have corrected this.

*13) Discussion, sixth paragraph: how can loss of mechanosensitivity in nociceptors (which is in any case no by no means complete) lead to noxious stimuli being perceived as innocuous? This speculation should be removed.*

We have removed this speculation.

*14) The Discussion should include more alternative models and not focus exclusively on membrane stiffness as the only explanation. Several important parts of the literature have been completely ignored.*

As described above we have modified the Discussion to make it more balanced.

*15) I cannot see whether the experiments measuring mechanosensitive currents in DRG neurons was done blind to the genotype and/or transfection. Please mention in the manuscript.*

We performed all experiments blind. This has been added to the manuscript.

*16) The last part of the Results section, mainly data presented in Figure 7, is less convincing. Figure 7 show acetylated tubulin in MEFs. I was very surprised by the image presented since similar experiments performed in other labs (using the same antibody) tend to only reveal the primary cilium. Why is the entire microtubular network visible here? Has the microtubule network been stabilized in some way (as was the case in another paper from the Heppenstall lab)? If so, it is not mentioned in the figure legend and/or Materials and methods section. Please add the missing information. This possible problem has no impact on the overall meaning of the figure (no sub-membrane localization of acetylated tubulin in MEFs) but it does need to be explained/corrected.*

The experiment was performed on serum starved MEFs to increase acetylation levels. This clarification has been added to the manuscript.

*17) The presence of a sub-cortical acetylated tubulin band in the cell body of DRG neurons (Figure 7) is very convincing. In contrast, data for saphenous nerve axons and sensory neuron axons are less convincing. To fully conclude, super-resolution imaging of axons will be needed, although I know this is hard to perform. To circumvent this requirement, I suggest that the authors use less affirmative phrasing in the Results/figure sections (subsection “Acetylated tubulin localizes to a prominent submembrane band in peripheral sensory neurons” and Figure 7 legend), that they provide higher-magnification images, or that they confine these results to the Supplemental data.*

We have reworded this section of the Results, discussed this aspect in the Discussion, and moved the appropriate figures to supplementary.

*18) The data relating to hyperosmotic experiments, presented in Figure 7, are not adequate; a before and after picture is required (as shown in the associated supplementary figure). In addition, the quantification indicated that 9 cells were analyzed in control conditions, but the cells shown in Figure 7 and in the supplementary data appear to be the same one. Could you provide another picture? Overall I think that the data relating to hyperosmotic shock, which is currently presented in the supplemental figure, should be included in Figure 7 in the main text.*

We have reorganized the figure and moved the raw images from supplementary to Figure 7.

*19) The Discussion is very interesting. I would strongly suggest that the authors tone down the sentence "We found that acetylated microtubules localize to a prominent band under the membrane of sensory neuron cell bodies and axons". For me, the presence of such a band in axons is not clearly demonstrated in the manuscript, even though I agree that it is likely to exist.*

We have corrected this statement.

*20) A recent publication established direct links between tubulin tyrosination and cellular stiffness in myocytes (Robison et al. 2016 Science). These results should be mentioned in the Discussion, and possible direct links between tubulin acetylation and detyrosinated microtubules should be discussed.*

We have added discussion on the Robison et al. 2016 Science paper.